# WORLD-MODEL BASED HIERARCHICAL PLANNING WITH SEMANTIC COMMUNICATIONS FOR AUTONOMOUS DRIVING

## ABSTRACT

World-model (WM) is a highly promising approach for training AI agents. However, in complex learning systems such as autonomous driving, AI agents interact with others in a dynamic environment and face significant challenges such as partial observability and non-stationarity. Inspired by how humans naturally solve complex tasks hierarchically and how human drivers share their intentions (e.g., using turn signals), we introduce HANSOME, a WM-based hierarchical planning with semantic communications framework. In HANSOME, semantic information, particularly text and compressed visual data, is generated and shared to improve two-level planning. HANSOME incorporates two important designs: 1) A hierarchical planning strategy, where the higher-level policy generates semantic intentions, and semantic alignment is devised to ensure that the lower-level policy determines specific controls to execute these intentions. 2) A cross-modal encoder-decoder to fuse and utilize shared semantic information and enhance planning through multi-modal understanding. A key advantage of HANSOME is that the generated intentions not only enhance the lower-level policy but also can be shared and understood by both humans and other AVs to improve their planning. Furthermore, we devise AdaSMO, an entropy-controlled adaptive scalarization method, to tackle multi-objective optimization problem in hierarchical learning. Extensive experiments show that HANSOME outperforms state-of-the-art WM-based methods in challenging driving tasks, enhancing overall traffic safety and efficiency.

## 1 INTRODUCTION

An ambitious goal of embodied AI is to develop cognitive agents capable of dynamically and adaptively planning to perform tasks in complex, high-dimensional environments. World-model (WM)-based reinforcement learning (RL), an end-to-end learning approach, has demonstrated significant potential. In WM, a latent dynamics model of the environment is first learned and then leveraged to train policies. However, applying WM to real-world applications, such as autonomous driving in traffic networks, presents numerous challenges. These environments involve heterogeneous agents interacting in environments with intertwined system dynamics. A key obstacle in such complex settings is insufficient information available to the ego agent, which operates under partial observability and must plan in non-stationary environments.

A promising solution to the above challenge is to enable agents to share information (Zhu et al., 2022) . Recent research has explored sharing different types of information, such as (encoded) partial observations (Jiang & Lu, 2018), hidden states (Sukhbaatar et al., 2016), policy and value networks (Peng et al., 2017), and (encoded) action intentions (Kim et al., 2020; Qi & Zhu, 2018). Intention sharing between vehicles has been demonstrated to be a practical and promising approach to improving safety and efficiency in real-world vehicle-to-vehicle (V2V) applications (Wang et al., 2023a; 2024; 2023b; Xie et al., 2021; Zhu et al., 2022).

However, it is nontrivial to ensure that the shared information can be understood and utilized by agents of interest, which is challenging in real-world applications, such as in mixed traffic where human drivers and different types of autonomous vehicles (AVs) co-exist. AVs may share sensor data (Yu et al., 2024; Xu et al., 2022a) or detection results (Xu et al., 2021), whereas human drivers tend to

share and interpret turn signals, text messages, or voice prompts from navigation apps. Moreover, human information sharing often takes place at the intention level, improving the communication efficiency. This also aligns well with the hierarchical nature of human thought processes. If AVs can understand and generate intentions like turn signals, texts, or voices, human drivers and AVs can communicate with ease. The end-to-end "black-box" approach, which maps observation inputs directly to actions such as steering and acceleration, impedes the sharing of interpretable intentions. In this work, we attempt to address this issue and answer the following question for WM-based RL for autonomous driving: "*How to generate interpretable information for semantic communications and utilize such information to improve planning among heterogeneous agents?*"

To address this question, we develop HANSOME, based on the key insight that humans can naturally solve complex tasks quickly by leveraging hierarchical thinking and decision-making (Wang et al., 2023c), an essential component of human intelligence. Specifically, the human brain possesses a structured architecture capable of not only controlling specific muscular patterns but also of planning more abstract goals (Turella et al., 2020). This approach further provides an avenue for efficient information sharing as discussed earlier.

With this insight, HANSOME features two important designs. The first is a hierarchical planning strategy that generates and shares text-based semantic intentions such as "right turn" and "left lane change", understandable by heterogeneous agents. The higher-level policy generates these intentions, and the lower-level policy determines concrete vehicle controls (e.g., acceleration and steering) to achieve the intentions. The second design is a cross-modal encoder-decoder, which fuses shared text-based intentions and visual information in the form of bird-eye-views (BEVs), into a latent representation for multi-modal understanding. The latent representation encapsulates rich information about the environment, surrounding vehicles, and historical context, driving HANSOME's end-to-end decision-making across both levels. Notably, HANSOME does not mandate shared semantics as inputs, as it can independently predict and plan based on its own observations. However, semantic communication significantly enhances traffic safety and efficiency. By leveraging universally understandable semantics, HANSOME is well-suited for heterogeneous agents with different underlying policies, seamlessly functioning in both standalone and cooperative modes. Further details are provided in Section 3.

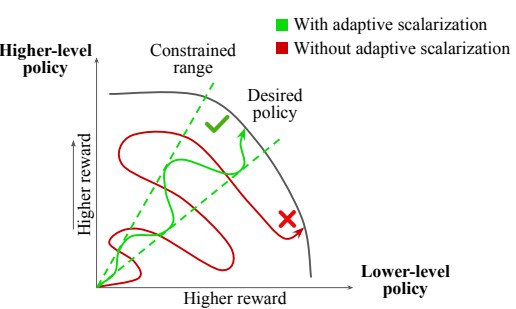

Figure 1: Illustration of AdaSMO for training hierarchical planning: The training of the two-level policies is essentially a two-objective optimization problem. Our AdaSMO method uses entropy-controlled adaptive scalarization to smooth out the oscillation between the two levels and accelerate the convergence to the desired policy.

Note that HANSOME's higher-level policy is not a replacement for route planning in Google Maps but an enhancement that leverages real-time perception to address immediate and complex decisions, given the rough route planned by map topology. While Google Maps can help avoid long-term routes with traffic jams by collecting user data (Mishra et al., 2018), this data is often delayed and does not account for real-time situations around the ego vehicle, such as sudden accidents or obstacles. Consequently, such route planning cannot make timely decisions, like determining whether to change lanes immediately or bypass an accident ahead. HANSOME bridges this gap by complementing map applications with a higher-level policy that integrates real-time perception for more dynamic and responsive decision-making.

A key challenge in training hierarchical planners is non-stationarity, as both policies evolve simultaneously. For example, the higher-level policy observes different transitions and rewards because the lower-level policy constantly changes, even in the same state with the same higher-level goal. This is a known challenge in hierarchical RL (Pateria et al., 2021; Hutsebaut-Buysse et al., 2022). Prior works usually mitigate the issue by updating transition data through relabelling and hindsight replay (Nachum et al., 2018; Levy et al., 2017; Jiang et al., 2019). Instead of relying on additional relabelling processes, we devise AdaSMO, an entropy-controlled adaptive scalarization technique, to train hierarchical planners. We view two-level training as a multi-objective optimization problem, which in general has a set of Pareto optimal points forming a Pareto frontier (as illustrated in Fig-

ure 1). A naive scalarization of the two objectives may yield poor results since the learned policy may oscillate across Pareto optimal points. With this insight, AdaSMO dynamically adapts the relative weights between the two policies, balancing their co-evolution by adjusting action entropy to control policy exploration, ultimately guiding them to converge on the desired policy.

Our main contributions are as follows:

- **HANSOME Design.** We introduce HANSOME, a WM-based hierarchical planning with semantic communications framework, to enable interpretable information sharing among heterogeneous agents. HANSOME has a *hierarchical planning strategy* where the higher-level policy generates and shares semantic intentions in the form of text to guide the lower-level policy which in turn decides specific controls. A cross-modal encoder-decoder is devised to fuse and understand the shared semantic information. Since information such as vehicle location or speed is not accessible in the WM's latent representation, we propose translating intentions into waypoints to enforce semantic alignment between higher-level intentions and lower-level controls. The reward function is meticulously designed to balance the objectives of intention generation and waypoint following.

- **Adaptive Scalarization in Multi-objective Optimization (AdaSMO) for HANSOME.** We view hierarchical training as multi-objective optimization and devise AdaSMO to dynamically balance learning of two-level policies to address non-stationarity. As the lower-level policy becomes more skilled, the higher-level policy progressively reduces its exploration by controlling action entropy, while gradually increasing the complexity of the lower-level subtasks.

- **Extensive Experiments on Complex Urban Driving Tasks.** We present extensive empirical results in Section 4 to demonstrate the capability of HANSOME on a variety of challenging urban driving tasks involving communications with other agents. Ablation studies are used to demonstrate the necessity of HANSOME's semantic communications and hierarchical planning in solving tasks where current state-of-the-art WM-based RL methods may fail, and show AdaSMO's effectiveness in training a good hierarchical planning strategy. Unlike prior WM-based RL works, HANSOME enables semantic communications across agents, and learns to generate and understand messages within WMs' imagination.

## 2 RELATED WORK

**World Models for Autonomous Driving.** WM studies in the field of autonomous driving can be grouped into two categories (Guan et al., 2024; Zhu et al., 2024). The first category leverages WMs as neural driving simulators to synthesize realistic driving videos (Yang et al., 2024; Li et al., 2023; Kim et al., 2021). For instance, GAIA-1 (Hu et al., 2023) generates driving scenarios from videos, texts, and actions. DriveDreamer (Wang et al., 2023d) and DriveDreamer-2 (Zhao et al., 2024) enhance scenario generation with high-definition maps and 3D bounding boxes, and integrate large language models for user-friendly interaction, respectively. ADriver-1 (Jia et al., 2023) advances this approach by eliminating the need for extensive prior information and achieving sustained driving through continuous scenario and action prediction. The second category utilizes WMs to train and evaluate agent policies within simulated environments. MILE (Hu et al., 2022) employs a Dreamer-style WM for imitation learning, utilizing road map and camera inputs to predict transitions in future BEVs. Prior works also explored Dreamer-style models for online RL. SEM2 (Gao et al., 2022) utilizes DreamerV2 and decodes camera and LiDAR data into semantic BEVs. Think2Drive (Li et al., 2024) trains DreamerV3 with BEV inputs on CARLA Leaderboard scenarios. Notably, MILE, SEM2, and Think2Drive use pre-determined routes provided by CARLA map topology to guide the ego agent. While HANSOME agents can determine their own routes using intentions generated by the higher-level policy; moreover, HANSOME utilizes semantic communications to improve planning.

**Hierarchical Reinforcement Learning (HRL).** HRL decomposes long-horizon tasks into simpler subtasks (Parr & Russell, 1997; Dayan & Hinton, 1992; Sutton et al., 1999), by learning a higher-level policy that operates on larger time scales, which provides subtasks to a lower-level policy that selects primitive actions to achieve them. Many prior works determine subgoal spaces through subtask discovery (Pateria et al., 2021; Hamed et al., 2024; Yang et al., 2019). For instance, Director (Hafner et al., 2022) learns sub-goal spaces directly from high-dimensional image space. HIRO (Nachum et al., 2018) and HAC (Levy et al., 2017) mitigate non-stationarity in hierarchical training by employing relabelling techniques and hindsight replay. However, the discovered higher-level goals are often not interpretable by heterogeneous agents, limiting their applicability in real-world multi-agent

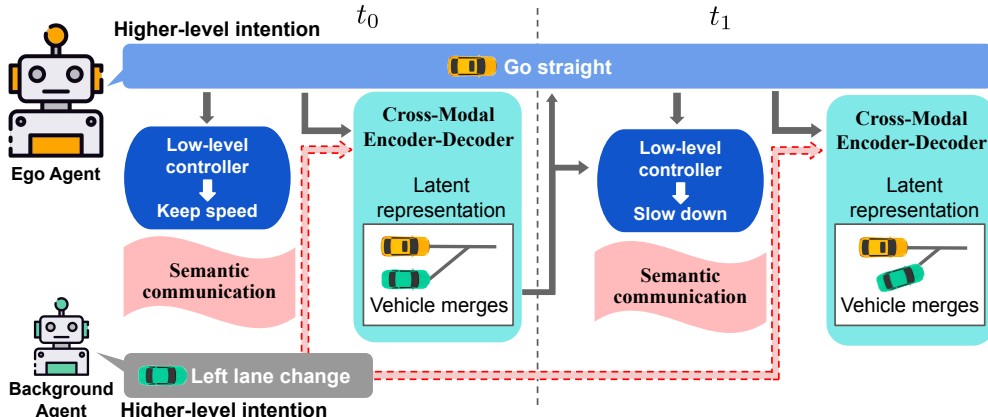

Figure 2: An example workflow of HANSOME: At time $t_0$, the higher-level policy of the background agent is to change to the left lane. Once this intention is shared with the ego agent, the ego agent predicts background agent's future trajectory through a cross-modal encoder-decoder in WM. In the next time step $t_1$, the ego agent slows down to avoid collisions upon detection of a trajectory crossing.

environments. HRL without subtask discovery generally requires domain knowledge to decompose tasks, using manually specified subtasks or semantic goal spaces, such as global XY coordinates for navigation (Andrychowicz et al., 2017; Nachum et al., 2018) or robot poses (Gehring et al., 2021). HAL (Jiang et al., 2019) uses language instructions as subgoals but is limited to single-agent object manipulation tasks and depends on relabelling. In contrast, our work advances hierarchical planning in mixed traffic environments, where agents communicate with others using understandable intentions generated by the planner. HANSOME does not require extra data relabelling to mitigate non-stationarity, instead employing AdaSMO to dynamically the adjust two-level learning.

**Information Sharing in Autonomous Driving.** Vehicle-to-vehicle (V2V) communications can significantly improve perception and, consequently, vehicle decision-making (Wang et al., 2018). The conventional approach focuses on sharing sensing information (Yurtsever et al., 2020) or trajectory sequences (Zhao et al., 2020; Han et al., 2019) with other agents, which can cause significant communication and computation overhead in complex real-world environments. Recent works have demonstrated the potential of intentional sharing to enhance traffic safety and efficiency in V2V applications (Wang et al., 2024; Xie et al., 2021). However, existing appraoches typically define intentions using GPS and vehicle heading. HANSOME takes a fundamentally different approach by introducing simple, interpretable text-based intention messages that are agnostic to *specific* sensor types or coordinate systems. In the related field of cooperative perception (CP), researchers have explored sharing raw sensor data (Yu et al., 2024; Xu et al., 2022a), intermediate features (Xu et al., 2022b), or detection results (Xu et al., 2021). While CP studies primarily focus on the perception module within modular pipelines and evaluate *open-loop* performance using metrics like segmentation and detection (Xu et al., 2022a), HANSOME distinguishes itself by implementing a *closed-loop* planner that directly interacts with realistic simulation environments, enabling more comprehensive evaluation of real-world performance. We provide further discussion for information sharing in multi-agent RL in Appendix A.

## 3 HIERARCHICAL PLANNING WITH SEMANTIC COMMUNICATIONS

To get a more concrete sense of HANSOME, we use an example to illustrate HANSOME's workflow.

*Example:* As illustrated in Figure 2, we consider two agents, where each agent has a *hierarchical planning strategy* that is capable of generating higher-level intentions in the form of texts and lower-level controls (e.g., acceleration, steering). Now, the background agent intends to change to the left lane, and this higher-level intention is shared with the ego agent. The *cross-modal encoder-decoder* of the ego agent, in turn, predicts the background agent's future trajectory using this shared intention. The generated low-dimensional latent representation is then used for next step planning. Consequently, the ego agent will slow down to avoid collisions.

Now, we proceed to provide a brief description of the world model design of HANSOME.

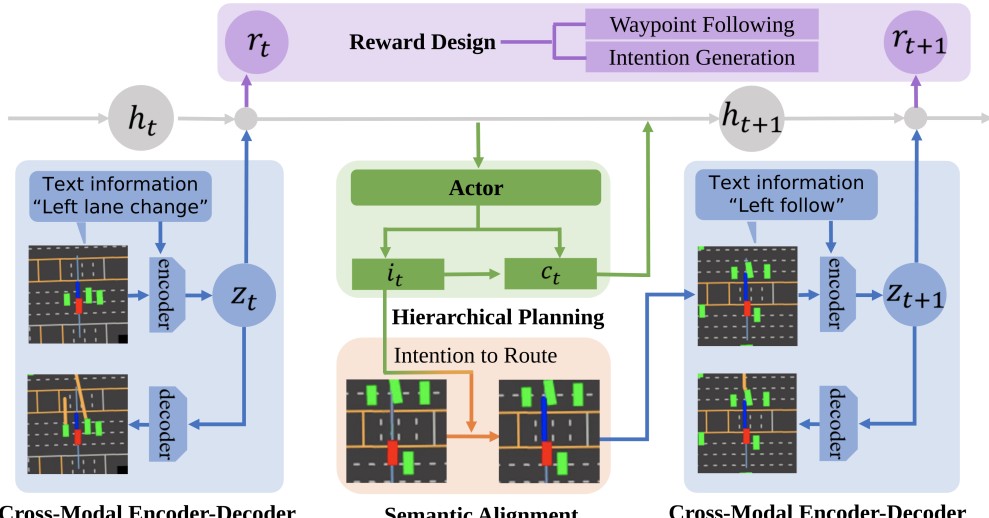

Figure 3: The structure of HANSOME. HANSOME consists of four key components: 1) Hierarchical Planning, 2) Semantic Alignment, 3) Reward Design, 4) Cross-Modal Encoder-Decoder.

**World Model.** We adopt the Dreamer-style WM paradigm to learn environment representations and dynamics through interaction (Hafner et al., 2023). The hierarchical policy is trained from scratch within the WM's imagination. The WM maintains an internal state $h_t$ using a Recurrent State Space Model (RSSM) (Hafner et al., 2020; 2023), which compresses the observations and actions from the past $t-1$ steps. Let $\phi$ denote the combined parameter vector of the WM. At each time step $t$, given the hidden state $h_t$, an encoder processes the observation $o_t$ ( (e.g., BEVs, destination, shared intentions) into a latent representation $z_t$, such that $z_t \sim p_\phi(z_t|h_t, o_t)$. Additionally, a dynamics predictor estimates $z_t$ without relying on $o_t$, i.e., $\hat{z}_t \sim p_\phi(\hat{z}_t|h_t)$. The model state is then defined as $x_t = [h_t, z_t]$. From $x_t$, the WM decodes an observation $o'_t \sim p_\phi(o'_t|x_t)$, predicts a reward $r_t \sim p_\phi(r_t|x_t)$, and estimates a discount factor $\gamma_t \sim p_\phi(\gamma_t|x_t)$, which represents the terminal probability. An actor is trained to generate actions $a_t$ conditioned on $x_t$. The RSSM then updates the internal state for the next time step as $h_{t+1} = f_\phi(x_t, a_t)$. HANSOME employs the Dreamer paradigm in its experiments; however, its design is not restricted to this specific structure and can easily incorporate future advancements in WMs.

We will present the details of four key components in HANSOME as illustrated in Figure 3, and then introduce AdaSMO to train the hierarchical policy in HANSOME.

### 3.1 KEY DESIGN COMPONENTS IN HANSOME

**Hierarchical Planning Aided by Semantics.** Human drivers naturally decompose driving maneuvers into subgoals. Thus inspired, HANSOME breaks down complex driving tasks into a series of semantic intentions that can be described using texts understandable by human drivers. The semantic information not only improves traffic safety and efficiency by informing other agents, but also guides the lower-level policy to achieve long horizon planning.

For the higher-level policy, we define a set of semantic intentions that align with human driving behaviors. This set, denoted as the intention space $I$, includes texts such as "Lane Follow", "Right Lane Change", and "Left Lane Change". We also denote the learned higher-level policy to be $\pi_\theta^H$. Every $T$ time steps, a new intention $i_t \sim \pi_\theta^H(\cdot|x_t) \in I$ is selected by the policy, conditioned on the current model state $x_t$. In this way, the complex task is decomposed into a sequence of subgoals in the form of semantic intentions.

Given a higher-level intention $i_t$, the lower-level policy $\pi_\theta^L$ maps the current model state $x_t$ to a control command $c_t$ (acceleration and steering), conditioned on $i_t$, i.e., $c_t \sim \pi_\theta^L(\cdot|x_t, i_t)$. Then the joint action can be represented as $a_t = [i_t, c_t]$, which is fed into the sequence model of the WM to predict the next frame. Theoretically, the dynamics of the environment depend only on $c_t$, but we include $i_t$ in the action to enforce semantic alignment for the lower-level policy, as elaborated below.

**Semantic Alignment for Hierarchical Planning.** Learning a lower-level policy that can effectively "understand" text-based intentions and align with their semantics is highly non-trivial, particularly in WM settings. This challenge arises because information such as vehicle location or speed is inaccessible in the WM's latent representation, making it impossible to directly evaluate alignment from the latent space. Since only the WM understands its own latent representations and can predict future rewards based on them, alignment can be reinforced through reward signals. Rewarding engineering has been the main challenge in RL, especially for complex task domains like autonomous driving (Kiran et al., 2021; Zhang et al., 2021). Designing a separate reward for each intention is cumbersome and impractical, as it does not scale with new intentions.

Therefore, we propose to visualize intentions to waypoints for semantic alignment. The semantics of the higher-level intention $i_t$ is "translated" into a sequence of waypoints $w_t = \{w_{t,i}\}_{i=1}^n$ that can be rendered on the BEV. The lower-level policy's objective is now to follow these waypoints on BEVs which implicitly aligns with the semantics. Since the translated waypoints exist in the observation space, the higher-level intentions must be included in the action space for the WM to accurately predict these waypoints. We emphasize in HANSOME, waypoints are planned by HANSOME itself, instead of being pre-determined by simulator as in Gao et al. (2022); Li et al. (2024).

**Reward Design for Hierarchical Planning.** Designing reward functions is known to be challenging in general. In HANSOME, reward design needs to take into account the signals at both levels, namely (1) generating intentions and (2) following waypoints. Our extensive experiments reveal that directly using a weighted sum of the two yields poor results. If the weight for (1) is too small, the higher-level policy may fail to learn the desired behavior. Conversely, a larger weight for (1) may overwhelm and disrupt the reward signal for learning the lower-level policy, thus significantly slowing down the training process. To overcome this challenge, we propose combining these components by dividing the waypoint-following reward by a factor proportional to the deviation extent of the intention from the overall destination. In this way, we can effectively amplify the impact of (1), while still providing a proportionate reward for the lower-level policy to follow the waypoints. Specifically, the reward for following the waypoints is given by

$$r_{\text{wpt}} = \alpha n + \beta v_\parallel - \gamma v_\perp - \kappa \mathbb{I}_{\text{collision}}, \tag{1}$$

where the first term represents the reward for reaching a waypoint and $n$ is the number of newly reached waypoints. The second term rewards the speed parallel to the route ($v_\parallel$) and the third term penalizes the perpendicular speed ($v_\perp$), which can effectively lead to a smoother trajectory. The last term is the penalty for collision. $\alpha$, $\beta$, $\gamma$ and $\kappa$ are scaling factors. Then, the complete reward function can be written as

$$r = r_{\text{wpt}}/(1 + A \cdot d_{\text{deviation}}) - B \cdot \mathbb{I}_{\text{invalid intention}} + C \cdot \mathbb{I}_{\text{reach destination}} \tag{2}$$

where $d_{\text{deviation}}$ represents the distance from the route planned by the higher-level policy to the overall destination. The last two terms penalize invalid planned routes and reward for reaching the destination, respectively. $A$, $B$ and $C$ are again scaling factors.

**Cross-Modal Encoder-Decoder.** The decoder in conventional WMs learns to reconstruct the input of the encoder by minimizing the MSE loss $\frac{1}{2}(p_\phi(o'_t|x_t) - o_t)^2$. In contrast, HANSOME adopts a cross-modal method to help WM fuse and "understand" the multi-modal semantic information. The cross-modal encoder-decoder takes BEVs and intentions shared by neighboring vehicles as inputs. Here, we assume that visual information is shared among neighboring vehicles, allowing the input BEV to combine this data and enhance observability. Intention information, rendered as waypoints alongside the destination directions, is incorporated into the BEV. The WM predicts the future trajectories of neighboring vehicles based on their shared intentions. As illustrated in Figure 3, the decoder output $o'_t$ includes bold orange lines representing the possible trajectories of background vehicles. Consequently, the new decoder loss is defined as:

$$\mathcal{L}_{\text{decoder}} \doteq \frac{1}{2}(p_\phi(o'_t|x_t) - o_t^m)^2, \quad o_t^m \doteq o_t \bigcup_{j \in \text{neighbors}} w_t^j, \quad w_t^j \doteq \{\text{location}_{t+i}^j\}_{i=1}^K. \tag{3}$$

Here, $w_t^j$ is the future trajectory of vehicle $j$, defined as the set of $K$ future locations. The input BEV, $o_t^m$, includes these trajectories rendered onto it. The decoder loss is the MSE between the decoder's output and $o_t^m$. By training the cross-modal encoder-decoder in this manner, trajectory information is effectively encoded into a unified latent representation, enabling its use for hierarchical planning.

Note that shared intentions are not obligatory inputs to HANSOME. When intentions form neighboring vehicles are absent, the encoder-decoder is trained to predict their trajectories based on history

movements. This makes HANSOME practical in real-world traffic systems, where heterogeneous agents coexist and some are unable to generate or communicate such information.

### 3.2 Learning Hierarchical Planning in HANSOME: A Multi-Objective Optimization View

Learning multiple levels of policies simultaneously is highly non-trivial due to the challenge of non-stationarity (Pateria et al., 2021). This is because the lower-level policy is non-stationary during training—even when given the same subgoal—so the trajectory it produces varies over time. This complicates the higher-level policy's learning process, as it observes inconsistent trajectories for the same subgoals. A classic approach to address non-stationarity is subgoal relabeling and hindsight replay (Andrychowicz et al., 2017; Levy et al., 2017; Jiang et al., 2019), where achieved states are used to relabel subgoals in transition data. Instead of relabeling transition data, we propose AdaSMO to dynamically adjust the higher-level policy exploration, thus balancing two-level learning, and mitigating non-stationarity without the need for data relabeling.

We view hierarchical policy learning as a multi-objective optimization problem with the two objectives of maximizing the reward of the higher-level and lower-level policy. For multi-objective optimization problems, there are a set of Pareto optimal points forming a frontier, as illustrated in Figure 1. The desired point on this frontier would enable the higher-level policy to plan a suitable route and the lower-level policy to execute it successfully. However, a naive scalarization of the two objectives may yield poor returns. Since there is no universal global optimum, the learned policy may oscillate between extreme points or converge to an undesired one. In our empirical studies, we observe that the higher-level policy converges much faster than the lower-level policy. When the lower-level policy is inadequate, the higher-level policy attempts to maximize rewards by generating overly simplistic plans, such as straight lines. As a result, the lower-level policy becomes fixated on these basic tasks and is unable to handle more complex ones.

**Entropy-Controlled Adaptive Scalarization.** To resolve these issues, we propose an entropy-controlled adaptive scalarization technique for multi-objective optimization (AdaSMO) to balance the training of higher-level and lower-level policies. The entropy of the higher-level policy is dynamically adjusted to embody different weights in the scalarization of the two objectives. A large entropy generates a nearly uniform distribution over the output intentions, in which case the primary goal is to enable the lower-level policy to learn to follow each individual intention. As the entropy decays, the higher-level policy reduces its exploration and begins to converge at a controllable rate toward the desired Pareto optimal. AdaSMO, therefore, echoes the human learning process, which begins with mastering basic skills before progressively integrating them into more complex tasks. Naturally, reward signals are used as a measure of policy quality to guide this adaptation. In practice, the entropy is adjusted by dividing the output of the higher-level policy's MLP head by a scaling factor $S$ before applying the softmax layer, i.e.,

$$p_\phi(i_t|x_t) = \mathrm{softmax}\left(\mathrm{MLP}(x_t)/S\right). \tag{4}$$

Let $\{a_1, \cdots, a_n\}$ be the output of MLP. The entropy of the higher-level policy will be

$$H(p_\phi(i_t|x_t)) = \ln\left(\sum_{i=1}^n \mathrm{e}^{a_i/S}\right) - \left(\sum_{i=1}^n \frac{a_i}{S}\mathrm{e}^{a_i/S}\right)\Big/\left(\sum_{i=1}^n \mathrm{e}^{a_i/S}\right), \tag{5}$$

which increases with $S$. Since the adjustment of $S$ depends on policy quality and is inherently task-specific, it is adjusted heuristically based on the average reward over the most recent $P$ episodes. As policies improve and rewards surpass certain thresholds, $S$ gradually diminishes to 1. Consequently, the algorithm initially prioritizes training a reasonably good lower-level policy, and then learning the higher-level policy by leveraging the enhanced lower-level policy. The relative performance trade-off can be controlled by the decreasing rate of $S$. In addition to $S$, we adjust other parameters, such as traffic density and the intention horizon $T$, to increase task difficulty along training process. For example, we start with a larger time window ($T = 128$ time steps) to allow the lower-level policy sufficient exploration, and then gradually reduce to $T = 1$ as the lower-level policy becomes more skilled and is allowed to change its intentions actively every time step. Notably, unlike prior hierarchical WM that uses a constant time horizon for high-level goals (Hafner et al., 2022), AdaSMO allows HANSOME to adjust the goal horizon dynamically based on the agent's skill level. More explanations and implementation details of AdaSMO are shown in Appendix D.1.

Table 1: Comparison between HANSOME and baseline algorithms in `DenseTraffic`. (Sum of success rate and collision rate is not equal to one since there are other cases such as time out.)

| Algorithms | Success Rate | Norm. Speed | Collision Rate |
|---|---|---|---|
| DreamerV2-C | 48.89 % ± 4.44% | 0.80 ± 0.05 | 51.11 % ± 4.43 % |
| Director-C | 66.67 % ± 6.67 % | 0.56 ± 0.01 | 33.33% ± 5.69 % |
| DreamerV3-C | 40.66% ± 2.78 % | 0.69 ± 0.02 | 51.65% ± 5.40 % |
| **HANSOME** | **88.17% ± 1.08 %** | **0.86 ± 0.15** | **3.03% ± 3.03%** |

## 4 EXPERIMENTS

In this section, we present extensive experiments to demonstrate the capabilities of HANSOME. In Section 4.1, we highlight the performance gains offered by the hierarchical planning and semantic communications in HANSOME compared with state-of-the-art WM-based approaches. In Section 4.2, we study the impact of semantic communications on improving the traffic efficiency and safety. Section 4.3 details the advantages of HANSOME's hierarchical planning for complex tasks. In Section 4.4, we highlight the benefits of using AdaSMO in HANSOME by comparing it with non-adaptive training.

**Benchmark Settings.** Dreamer-style and online RL works typically evaluate models in a highly realistic simulator, CARLA, with customized scenarios (Gao et al., 2022; Pan et al., 2022; Xie et al., 2021) or Leaderboard (Li et al., 2024), given the interactions with environments needed by online RL and its closed-loop nature. See Appendix A for how prior works customize tasks to their needs. Therefore, to examine the benefits of semantic communications in HANSOME and the baselines, as well as the advantages of the hierarchical planner, we develop four challenging tasks to evaluate various model capabilities, including 1) `DenseTraffic`, a task featuring dense traffic with 300 randomly spawned vehicles in CARLA *Town04*. The ego agent needs to navigate through traffic flows, change lanes, and avoid collisions to reach destinations. 2) `LeftTurn` and 3) `RightTurn` are tasks performed at intersections, where the agent has to merge into dense traffic at the proper time when other vehicles randomly divert from the flow. Notably, the background vehicles are set to be aggressive such that they do not actively avoid collisions, mimicking irrational human drivers. 4) `ObstacleBypass`, where the agent is asked to go straight on a lane but with an obstacle ahead. It tests the agent's flexibility to deviate from the pre-determined route destination and return later.

**Baseline Settings.** Our baselines include state-of-the-art WMs, DreamerV2 (Hafner et al., 2020), Director (Hafner et al., 2022), and DreamerV3 (Hafner et al., 2023). Director generates images as higher-level goals but does not present actionable text semantics; DreamerV2 and DreamerV3 are single-level planning frameworks. Prior applications of WM-based agents in CARLA (Gao et al., 2022; Hu et al., 2022; Li et al., 2024) typically adopt Dreamer-style single-level planning. For example, Think2Drive (Li et al., 2024) uses DreamerV3 with BEV inputs; SEM2 (Gao et al., 2022) modifies DreamerV2 decoder to output BEV masks. Both models are Dreamer-based and not open-sourced; therefore, we use DreamerV2 and DreamerV3 to represent them. These baselines do not take into account multi-agent interactions. There is not yet a hierarchical or communicative WM-based RL algorithm as our baselines, nor closed-loop autonomous driving benchmark with communications as benchmarks. For fair comparison, we assume enhanced observability via semantic communications across all baselines. To highlight this, we denote baselines with a "-C" suffix (see Table 1) meaning semantic communications are enabled for these baselines.

### 4.1 OVERALL PERFORMANCE OF HANSOME

Firstly, we compare HANSOME and three baselines on `DenseTraffic`. All agents receive BEVs with enhanced observability via online interactions with other agents. Baseline agents determine their controls by looking at pre-determined routes rendered on BEVs, a common practice in the community (Li et al., 2024; Hu et al., 2022). An episode terminates upon collision, going out of lane, and time out. For HANSOME, the task is even more challenging in that the agent only takes in a destination point and has to plan its own routes towards the destination, and the termination is additionally triggered by invalid higher-level plans.

We use success rate, collision rate, and normalized speed with respect to the desired speed to measure the ego agent's safety and efficiency. The comparison in Table 1 corroborates that HANSOME

Table 2: Comparison between the BEVs imagined by WM and the ground truth. The background vehicle's future trajectory (the bold orange lines in BEVs) is not available in inputs to the WM; WM predicts them using cross-modal encoder-decoder to fuse shared text-based intentions into BEVs.

| Time | 1 | 7 | 22 | 29 | 33 | 36 | 46 | 51 |
|---|---|---|---|---|---|---|---|---|
| Imagined BEVs | | | | | | | | |
| Ground Truth | | | | | | | | |

Table 3: Comparison of different communication settings for `LeftTurn` and `RightTurn`.

| | LeftTurn | | | RightTurn | | |
|---|---|---|---|---|---|---|
| | Collision Rate | Norm. Speed | Success Rate | Collision Rate | Norm. Speed | Success Rate |
| w/ visual only | 16.94% ± 4.67% | 0.50 ± 0.01 | 82.21% ± 6.94% | 8.38% ± 3.04% | 0.64 ± 0.04 | 91.62% ± 3.04% |
| w/ visual + intention (HANSOME) | **13.89% ± 3.21%** | **0.60 ± 0.01** | **85.19% ± 4.14%** | **5.52% ± 0.45%** | **0.72± 0.07** | **94.27% ± 0.63%** |

achieves significantly better performance in all three metrics. Previous baseline algorithms are trained to follow fixed routes. They have to change lane if given that guidance at a fixed position regardless of whether there are other vehicles in that lane. Slowing down can avoid collisions but make it less efficient. HANSOME has the ability to actively re-plan to follow the current lane and change lane at the proper time, thanks to its hierarchical planning. The demos are shown in Figure 8 in the appendix.

## 4.2 ABLATION OF SEMANTIC COMMUNICATIONS

A key advantage of our approach is that HANSOME can predict and render the future trajectories of neighboring vehicles through the cross-modal encoder-decoder. Table 2 compares the WM's imagination of 64 future steps with the ground truth. In the first three columns, the WM accurately predicts the locations and future trajectories (bold orange lines in the figure) of neighboring vehicles. In the remaining columns, deviations occur in two BEVs, primarily involving vehicles that have not been previously observed. This behavior is reasonable, as the WM can only infer the presence of vehicles beyond the BEV's visible range. These deviations highlight the WM's generalization ability in imagining new and unseen scenarios for training policies.

To justify whether the ego agent can effectively leverage the enhanced predictions to achieve safer and more efficient maneuvering in complex traffic tasks, we evaluate HANSOME on `LeftTurn` and `RightTurn`. The tasks evaluate the ego agent's ability to predict the future trajectories of other vehicles in a dense traffic flow and find the proper time to merge into the flow. The background vehicles follow aggressive policies, simulating irrational drivers, which increases the task's difficulty and necessitates accurate predictions of other vehicles' future movements. We compare HANSOME

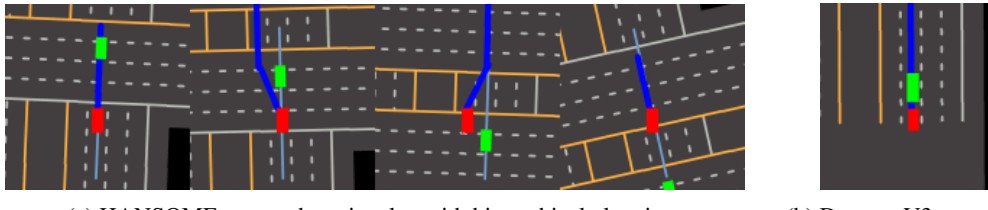

(a) HANSOME agent adapts its plan with hierarchical planning.  (b) DreamerV3 gets stuck.

Figure 4: Performance comparison between HANSOME and DreamerV3 when facing an obstacle.

against HANSOME without intention sharing. The metrics in Table 3 showcase that HANSOME significantly reduces collision rates by $18\% - 34\%$ via intention sharing and improves efficiency by $11\% - 20\%$ indicating more confident policies.

### 4.3 Ablation of Hierarchical Planning

The hierarchical planning capability of HANSOME allows it to re-plan and navigate flexibly and efficiently through complex traffic scenarios. This is partly demonstrated in Section 4.1. Here, we evaluate HANSOME on `ObstacleBypass` to further highlight its advantages. Figure 4a showcases how the agent smartly re-plans to deviate from its destination lane and swiftly merges back. For comparison, we trained a DreamerV3 agent to follow the planned route on BEVs, similar to SEM2 (Gao et al., 2022) and Think2Drive (Li et al., 2024). Since such agents are trained to follow routes given by static CARLA map topology instead of planning their own routes, they find it hard to initiate temporary deviation from the original route and return later, as demonstrated by the agent standing still in front of the obstacle in Figure 4b. See Figure 8 in the appendix for another example where HANSOME actively re-plans to avoid collisions.

### 4.4 Ablation of AdaSMO

We propose AdaSMO to address the challenges introduced by multi-objective optimization, specifically the oscillation between Pareto-optimal points during hierarchical policy training. In this ablation study, we compare the training performance of AdaSMO with the vanilla multi-objective approach. As shown in Figure 5, the vanilla approach converges prematurely to a local optimum at approximately 100K training steps, while AdaSMO continues to learn and achieves a significantly higher reward. This im-

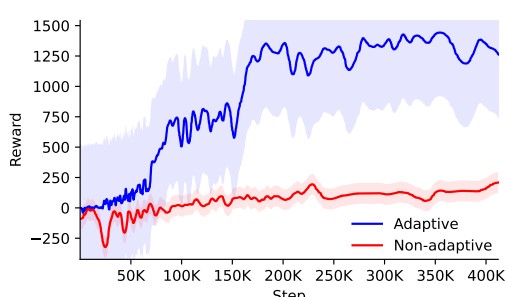

Figure 5: Entropy-controlled adaptive scalarization vs. vanilla scalarization (non-adaptive)

provement is attributed to AdaSMO's strategy of prioritizing lower-level policy learning while enabling extensive exploration of the higher-level policy during initial training. Exploration gradually decays as the lower-level policy stabilizes. Sharp increases in reward at around 70K and 150K steps highlight AdaSMO's effectiveness in adjusting the action entropy to optimize learning dynamics.

## 5 Discussion

We propose HANSOME, a WM-based hierarchical planning framework that mirrors the human approach of decomposing driving behaviors into different levels of abstraction and using turn signals to inform other drivers. HANSOME seamlessly integrates hierarchical planning with semantic communication using visual and textual information. The higher-level policy generates text-based intentions to guide the lower-level policy and to communicate with other agents. HANSOME employs a novel adaptive approach, AdaSMO, to tackle the challenging multi-objective optimization in hierarchical planning. Through WM-based RL, HANSOME learns both higher-level and lower-level policies from scratch within the WM's imagination, mastering complex driving tasks and effectively navigating dense traffic. Extensive experiments demonstrate that HANSOME outperforms state-of-the-art WM-based RL algorithms and enhances traffic safety and efficiency through its hierarchical planning and semantic communication capabilities.

We further discuss our ego-centric training algorithm in Appendix B. Multi-agent RL is notoriously difficult to train due to intertwined dynamics (e.g., when poor initial policies send inconsistent messages that misaligned with the agents' behaviors), making it difficult for others to learn effectively from them. However, by utilizing ego-centric learning with semantic communications, our experiments show that HANSOME successfully learns during training and generalizes to multi-agent execution to some extent. We observe that multiple HANSOME agents interact and negotiate with other HANSOME agents or rule-based agents through semantic communications in mixed-agent high-dimensional environments. While HANSOME demonstrates promising potential in multi-agent tasks, future studies are needed to investigate the multi-agent RL training.

**Ethics Statement**    Our research presents HANSOME, a world-model-based hierarchical planning framework that integrates semantic communication to enhance autonomous driving in mixed traffic environments. In line with the ICLR Code of Ethics, we have considered the ethical implications of our work. Our research does not involve human subjects or sensitive personal data; all models are trained from scratch using in CARLA simulation. By adhering to ethical principles, we aim to contribute positively to autonomous driving, fostering advancements that are socially responsible and beneficial to society.

**Reproducibility Statement**    We ensure reproducibility by submitting the source code of HANSOME as supplementary materials. The source code provides the model implementation, the CARLA benchmark implementation, and all the model hyperparameters and task configurations needed to reproduce the results shown in the paper. Instructions for running training and evaluation are also included in our code's documentation. The model settings and hyperparameters are presented in Appendix F; AdaSMO training and evaluation processes are discussed in Appendix D; the task configurations for CARLA simulation and all task benchmarks are showcased in Appendix E.

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

# Appendix

## A   FURTHER DISCUSSIONS

**Customized CARLA Benchmarks.**   It is a common practice for WM-based online RL to customize CARLA scenarios to test their algorithms. SEM2 (Gao et al., 2022) and Iso-Dream (Pan et al., 2022) uses a task in *Town03* to let the agent maximize rewards within 1000 steps and avoid collisions along the way. SEM2 uses 100 vehicles, and Iso-Dream uses 20 vehicles for training and 10 vehicles for testing. LILI (Xie et al., 2021) customizes a task where ego vehicle has to avoid an aggressive vehicle when moving forward to verify its opponent modelling.

**Comparison with Multi-Agent RL.**   Multi-agent RL (MARL) with communication has been extensively studied (Zhu et al., 2022), enabling RL agents to share past observations (Sukhbaatar et al., 2016), actions (Peng et al., 2017), or intentions (Kim et al., 2020). However, MARL often struggles with tasks such as autonomous driving, which take place in high-dimensional environments with complex dynamics. Additionally, MARL faces challenges due to the co-evolution of policies, which causes non-stationarity and hinders effective learning when agents interact with one another. To avoid these problems, we adopt ego-centric learning to enable policy learning with lightweight communication in multi-agent systems. We discuss details of ego-centric learning in Appendix B. Furthermore, our experiments demonstrate that, although HANSOME learns in an ego-centric manner, it generalizes to multi-agent scenarios that include a mix of HANSOME agents and rule-based agents to some extend.

**Comparison with Large Language Models (LLMs) for Autonomous Driving.**   Recent studies have explored the application of LLMs in autonomous driving (Yang et al., 2023), such as Driv-eLM (Sima et al., 2023), DriveVLM (Tian et al., 2024), Dilu (Wen et al., 2023), GPT-Driver (Mao et al., 2023), and DriveGPT4 (Xu et al., 2024). For instance, DriveLM and DriveVLM optimize natural language processing metrics by comparing scene descriptions and analyses with ground-truth annotations, such as driving captions or visual question answering, utilizing GPT-based models. Their "hierarchical planning" involves generating action descriptions from text prompts and converting these descriptions into waypoint tokens. However, this waypoint tokenization relies on trajectory statistics from training data, and there is no actual control to execute the plan in closed-loop settings. In contrast, LMDrive (Shao et al., 2024) is a closed-loop approach, where the higher-level instructions are provided as inputs for the vehicle to follow. PlanAgent (Zheng et al., 2024) introduces a chain-of-thought module to understand scenes and plan routes with text prompts. HANSOME's models are significantly more *lightweight*, comprising approximately 30 million parameters, enhancing its practicality for real-time inference. HANSOME learns both higher-level and lower-level policies from scratch within the WM's imagination, and evaluate policies in a *closed-loop* manner. Its higher-level policy generates semantic intentions without depending on prior knowledge from LLMs or trajectory statistics in datasets. Furthermore, LLM-based approaches typically focus on processing and understanding natural language inputs to reason and inform driving decisions, which may not encompass the full spectrum of data required for autonomous driving. HANSOME, however, integrates image inputs directly into its learning and decision-making process without intermediate text prompts and outputs, enabling a more holistic understanding of the driving environment and enhancing its ability to make informed decisions.

**End-to-End (E2E) Autonomous Driving and Benchmarks.**   Autonomous driving has witnessed rapid growth recently thanks to the advacement of E2E approaches (Chen et al., 2023). Unlike conventional approaches that employ a modular design and separate perception, prediction, planning modules, E2E approaches aim at producing driving plans or actions directly from raw sensor data inputs. Prior studies can be roughly categorized into two folds: imitation learning (IL)  (Chen et al., 2020; Prakash et al., 2020; Zhang & Cho, 2017; Shao et al., 2023; Chitta et al., 2022; Hu et al., 2018), and reinforcement learning (RL)  (Li et al., 2024; Gao et al., 2022; Zhang et al., 2021; Chekroun et al., 2023; Toromanoff et al., 2020; Zhang et al., 2022) methods. Several open-loop benchmarks were developed to test E2E approaches, including CARLA (Dosovitskiy et al., 2017), nuScenes (Caesar

et al., 2020), Argoverse (Chang et al., 2019), Waymo (Schwall et al., 2020), and nuPlan (Caesar et al., 2021). Recently, closed-loop benchmarks like CARLA have become more recommended for research (Chen et al., 2023), as there is no strong evidence to suggest that good open-loop results correlate with good closed-loop performance. Dreamer-style works (Gao et al., 2024; Li et al., 2024; Pan et al., 2022), due to their interactive demands for online RL, often use CARLA as a closed-loop benchmark. CARLA allows flexible control over environments and background traffic, which is essential for evaluating HANSOME, as it requires multi-agent interactions and semantic communications in complex, dense traffic scenarios.

**Comparison with Hierarchical Planner in Embodied AI.** Our related work discussion on hierarchical planning focuses on reinforcement learning, as it aligns with our approach. We want to further discuss the advancements in embodied AI community that enables the agent to plan over language abstractions, typically through Large Language Models (LLMs). HiP (Ajay et al., 2024) uses LLMs to construct symbolic plans, and trains a visual model, an action model, jointly to solve long-horizon tasks. VLP (Du et al., 2023) uses vision-language models as both policies and value functions; a text-to-video model is trained to generate video plans that illustrate how to complete the final task. Unlike these embodied AI works, HANSOME considers environments where heterogeneous agents communicate for better planning. Moreover, HANSOME is a lightweight online RL framework that does not rely on offline data or expert demonstrations. It has 30 million parameters, significantly fewer than large vision or language models in embodied AI research, which can be critical to fulfill low-latency demands of AVs.

**An Example of Multi-Objective Optimization View for HRL** In principle, the training of hierarchical planning can be viewed as a multi-objective optimization problem, with two objectives being to maximize the reward of the higher-level policy and the lower-level policy, and there are trade-offs between the two objectives in general. For instance, the higher-level policy may plan sophisticated routes involving frequent lane changes and overtaking maneuvers to reach the destination faster, without considering whether the lower-level policy can realistically execute such complex maneuvers, leading to poor lower-level performance. On the other extreme, a poor higher-level policy may adhere to simplistic plans like a straight-line path, so that the lower-level policy can achieve nearly perfect performance in following just a straight line.

## B Ego-centric Learning

A challenge to address for HANSOME is the source of shared intentions during training. A straightforward approach is to spawn multiple agents in the environment, each independently controlled by the hierarchical policy, and allow them to communicate with each other. However, the main drawback of this approach is that these agents will initially not follow their generated intentions due to the lack of a good lower-level policy. Since the WM needs to predict the trajectories of background vehicles based on their shared intentions, any misalignment between the agents' behavior and their intentions will mislead the WM in understanding these intentions. Only when the lower-level policy is sufficiently good at following intentions can the WM begin to learn the correct interpretation of each intention, enabling the agent to use this information for better planning.

To mitigate this issue and accelerate the training process, we use a distributed learning method to train HANSOME. In particular, the background vehicles that the agent interacts with are controlled by CARLA's autopilot (Dosovitskiy et al., 2017), a rule-based autonomous driving algorithm. Although not perfect, the autopilot can primarily follow a randomly generated route. Their intentions and planned routes can be extracted from CARLA's traffic manager and shared with our agents. In this way, the agent can simultaneously learn to follow the higher-level intentions it generates and utilize the shared intentions from other vehicles for better planning. Despite being trained in a distributed manner, our experiments demonstrate that the agents generalize well in multi-agent environments (see Figure 6).

**Generalization to Multi-agent Learning** Previous E2E autonomous driving works in CARLA (Li et al., 2024; Gao et al., 2021) train and evaluate the ego agent in environments where the background vehicles are controlled by rule-based CARLA autopilots that have privileged information to CARLA environments.

In this section, we are showing that through ego-centric learning and semantic communications, HANSOME agents can generalize to multi-agent environments where multiple HANSOME agents interact and share information to negotiate with others.

As shown in Figure 6 and Figure 7, we test the behaviors of two HANSOME agents, which are both trained in an ego-centric learning manner, but not in the multi-agent environment, when they meet and want to change to each other's lane. They illustrate interesting bargaining process for the priority of performing lane change. Specifically, they are spawned at the leftmost and rightmost lane respectively, and are required to change to each other's lane. When they meet at the middles lanes where their planned routes cross, the higher-level planners of both agents keep re-planning new trajectory to avoid possible collision and jam. Eventually, one agent slows down to make room for another and they successfully complete the task.

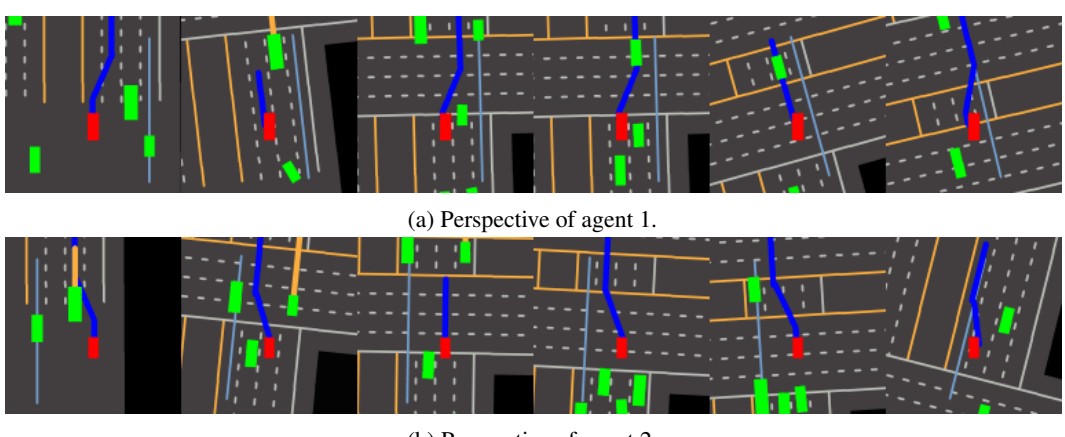

(a) Perspective of agent 1.

(b) Perspective of agent 2.

Figure 6: Two HANSOME agents interact with each other and background vehicles in the `DenseTraffic` task. They are both trained in an ego-centric learning manner, but not trained in the multi-agent environment. They are spawned at the leftmost and rightmost lane respectively, and are required to change to each other's lane. When they meet at the middles lanes where their planned routes cross, he higher-level planner of both agents keeps re-planning new trajectory to avoid possible collision and jam. Finally, they successfully complete the task.

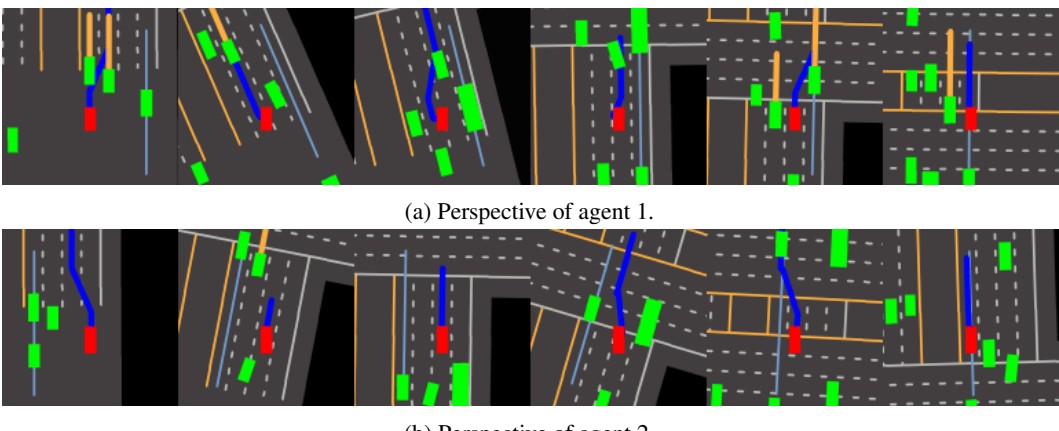

(a) Perspective of agent 1.

(b) Perspective of agent 2.

Figure 7: Another example of ego-centric learning agents interaction in `DenseTraffic` task.

## C  Driving Demos

**DenseTraffic**  Figure 8 shows a case where HANSOME re-plans to avoid the collision. In the first frame, it intends to change to the right lane. In the second frame, however, it detects a vehicle behind, so it cancels the plan and keeps going straight until the vehicle passes, allowing it to safely change the lane in the last two frames. Figure 9 shows some cases where agents without hierarchical planning fail.

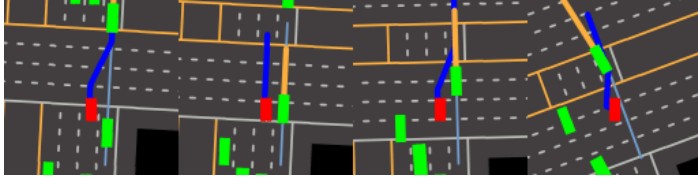

Figure 8: Hierarchical planning enables vehicle re-plan and avoid obstacles adaptively.

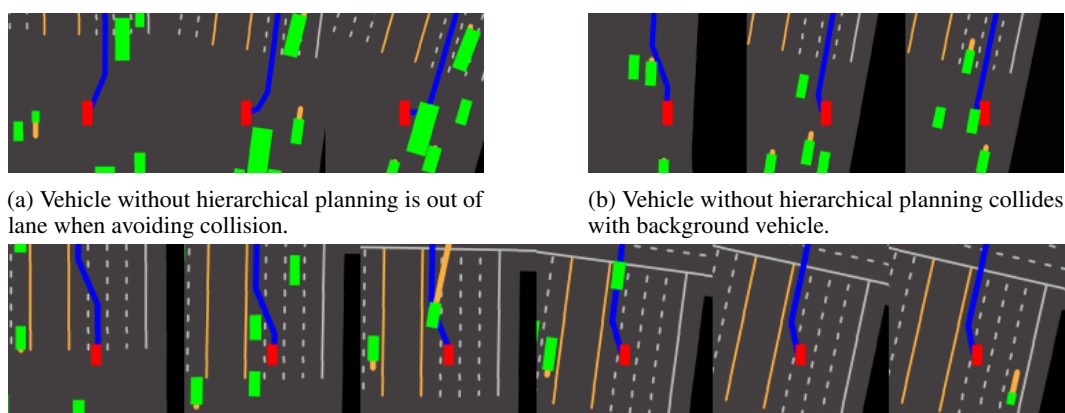

(a) Vehicle without hierarchical planning is out of lane when avoiding collision.

(b) Vehicle without hierarchical planning collides with background vehicle.

(c) Vehicle without hierarchical planning is not sure and keeps waiting for a long time when changing lane.

Figure 9: Examples of vehicle without hierarchical planning fails to deal with lane change.

**LeftTurn and RightTurn**  Figure 10 and Figure 11 show how semantic communications help the agent succeed in penetrating the car flows to make turns at the crossing. Taking Figure 10a as an example, the car flow is too dense for the ego agent to cut in. However, when the next vehicle shows the intention to turn right, making room for the ego agent, it successfully cuts into the flow without collision. In the case as Figure 10b when the intentions are not shared, the ego agent learns to predict the intentions of background traffic based on their behavior. Without explicit information about the intentions of background traffic, the ego agent learns a conservative policy, resulting in reduced safety and efficiency. Nonetheless, the ego agent can still cut into the flow and navigate through the intersection. This demonstrates that HANSOME is a robust framework that does not entirely depend on intention sharing, making it dependable in realistic environments.

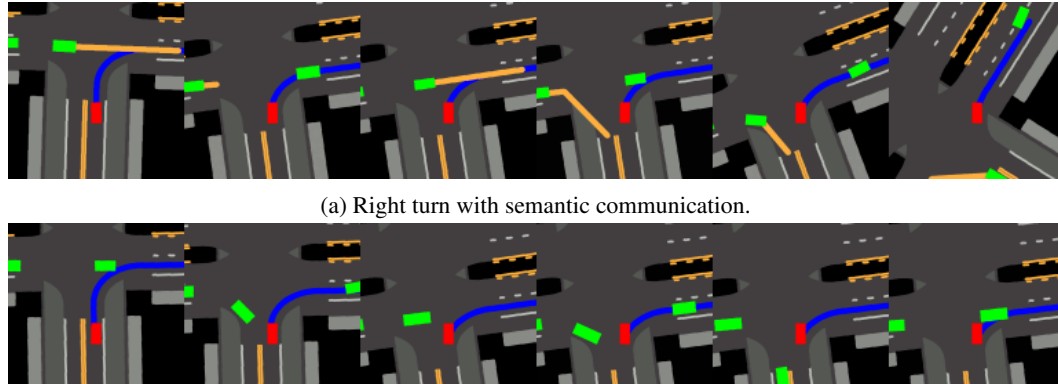

(a) Right turn with semantic communication.

(b) Right turn without semantic communication.

Figure 10: The HANSOME agent with semantic communication can find the proper timing to cut into the traffic flow when leading vehicle turns right and will not interfere with it, whereas the agent without semantic communication learns a conservative policy with a relative lower success rate to complete the task.

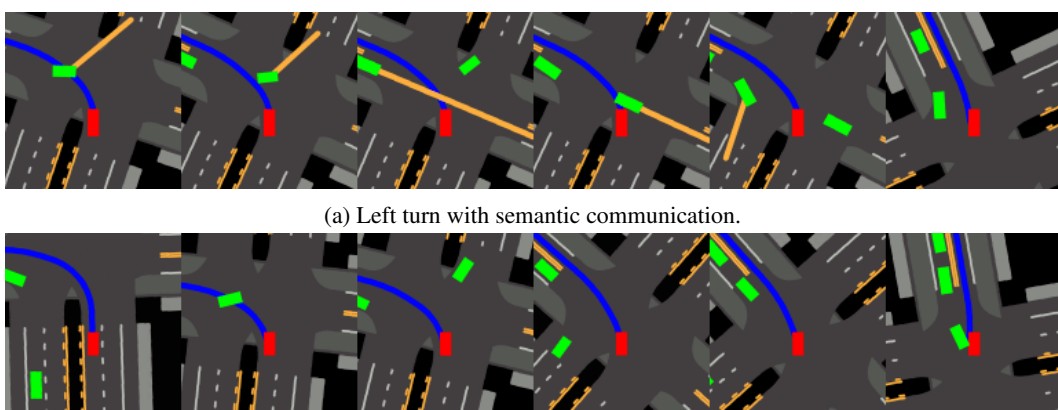

(a) Left turn with semantic communication.

(b) Left turn without semantic communication.

Figure 11: The HANSOME agent with semantic communication can find the proper timing to cut into the traffic flow based on shared intentions. In contrast, the agent without semantic communication has lower success rate and leads to collision occasionally.

## D    TRAINING & EVALUATION

Our baseline and HANSOME agents were trained on NVIDIA A100 GPUs. Each agent requires around 20 GB memory, including 3-4 GB for CARLA. Due to CARLA's GPU resource consumption, it takes around 15 GPU hours to reach 150k steps for our most challenging `DenseTraffic` task with 300 vehicles, and 15 GPU hours to reach 400k steps for other tasks with less traffic. For each task, we use the same training step budget for all HANSOME and all the baseline models.

The model is trained through an online manner where the agent has to learn from scratch without any expert demonstrations. The communication generation and understanding is also learned online.

To ensure fair comparison, we enable semantic communications of all baseline models, identical RSSM settings in WMs as shown in Appendix F, CARLA simulation and task configurations as shown in Appendix E.

For evaluation, we collected ego vehicle rollouts in the online environments for 300 episodes with 3 different random seeds to compute the performance metrics, and confidence intervals.

## D.1 ADASMO LEARNING

There are two parameters, unimix, and entropy scaling factor, that can be used to control entropy of the higher-level policy. We initially apply a unimix of 1.0 (equivalent to an infinite $S$) to allow the higher-level policy to conduct fully random exploration, enabling the lower-level to explore each possible intention sufficiently. Heuristically, when the overall rewards increase, $S$ becomes smaller and gradually reduces to 1.

**Adaptive Learning Processes.** AdaSMO adjusts higher-level policy exploration by accounting for the proficiency of the lower-level policy. This approach is inspired by the human learning process, which begins with mastering basic skills before progressively integrating them into more complex tasks. Naturally, reward signals are used as a measure of policy quality to guide this adaptation. We evaluate the lower-level policy's skill level through the average reward $\bar{R}$ over the recent $P$ episodes. Let $B_1, B_2, ..., B_n$ be the thresholds for certain $\bar{R}$. The entropy scaling factor $S$ is adjusted based on thresholds defined as follows:

$$
S(\bar{R}) = \begin{cases} S_\infty, & \text{if } \bar{R} \le B_1 \\ S_1, & \text{if } B_1 < \bar{R} \le B_2 \\ S_2, & \text{if } B_2 < \bar{R} \le B_3 \\ \vdots \\ S_n, & \text{if } \bar{R} > B_n \end{cases} \tag{6}
$$

Reward signals in reinforcement learning are inherently task-specific. AdaSMO can be viewed as an adaptive exploration adjustment mechanism, with its parameters determined by the nature of the task domain and the current policy quality. This concept is analogous to adaptive learning rate adjustment Liu et al. (2019), where learning rates are tailored to the datasets on which neural networks are trained, and current performance.

We present an example set of thresholds and parameter adjustments for each stage in Appendix D.1. It is important to note that AdaSMO is robust to variations in these parameters—just as different learning rate adjustment strategies can still yield optimal policies, albeit with varying convergence speeds. We will discuss this effect further in the context of the Warm-Up concept below.

**AdaSMO Warm-Up** We introduce the concept of warm-up, through which the lower-level policy is prioritized during training while the higher-level policy remains random exploration. It is critical to specify a proper timing to terminate the warm-up. We heuristically use the extrinsic reward to threshold the warm-up.

Specifically, we experimented with thresholds 80 and 100 to terminate the warm-up at 30K (red curve) and 70K (blue curve) steps, respectively. We notice that adjusting the warm-up termination timing has a significant influence on AdaSMO training speed. During warm-up, the higher-level policy keeps a high degree of exploration. This enables lower-level policy being trained to follow instructions from higher-level. If the warm-up is terminated too early, the lower-level policy has not been well trained for lane-following. Thus, the higher-level policy, whose training largely depends on lower-level policy performance, cannot improve immediately. The significant delay after warm-up termination and before reward curve rises up can be observed on red curve in Figure 12. The warm-up terminates at 30K steps while the reward grows only after 150K steps. In the contrast, a proper warm-up termination timing results in well trained lower-level policy. The higher-level policy can be trained based on a stable lower-level lane-following policy and improved rapidly, as the rapid growth at 70K steps shown on the blue curve. However, in both situations, the reward converges to the same level, which means our method is robust to the hyper-parameters, while a good set of parameters can significantly speed up training.

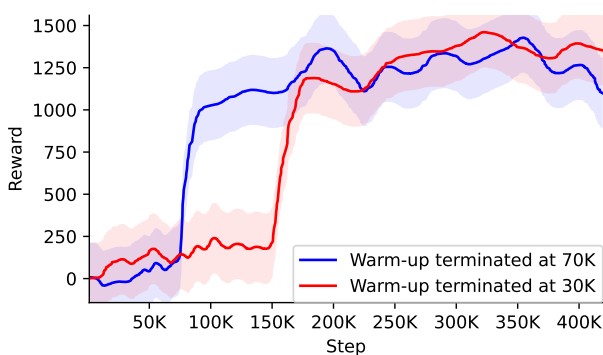

Figure 12: The effect of warm-up termination timing on AdaSMO training.

| Threshold | Adjusted Parameters |
|---|---|
| **Warm-up (Initial Parameters)** | |
| 0 | Unimix = 1.0
Horizon = 128
Entropy Scale = 1.0
Allow Replanning = False
Vehicles = 50 |
| **After 100 Reward** | |
| > 100 | Unimix = 0.0
Horizon = 16
Entropy Scale = 3.0
Allow Replanning = True
Vehicles = 300 |
| **After 120 Reward** | |
| > 120 | Entropy Scale = 1.5 |
| **After 250 Reward** | |
| > 250 | Vehicles = 300, Allow Replanning = True |
| **After 450 Reward** | |
| > 450 | Entropy Scale = 1.0 |

Table 4: Adaptive parameter adjustment based on average rewards.

## D.2 ACTION SPACE

We include the action space settings in Table 6. Here is a detailed explanation of the action space design.

**Intentions.** In CARLA map topology, there are six driving commands: three for movement on lanes (follow lanes, change to left lane, or change to right lane) and three for movements at intersections (go straight, turn left, or turn right). HANSOME uses a set of driving primitives in three directions as the higher-level intention space $I = Straight, Left, Right$. These intentions cover possible driving behaviors on both lanes and intersections. This intuitive setting also aligns with how human drivers use turn signals with three states (left . It is also a common practice in the AV community; for example, VAD (Jiang et al., 2023) uses going straight, left, or right as the 3-dimensional high-level action space. The high-level action of VAD is obtained through pre-determined routes, while HANSOME can generate intentions on its own.

**Vehicle Controls.** The vehicle controls given by the lower-level policy use a $5 \times 3$ dimensional one-hot encoding, where there are 5 discrete steering values and 3 acceleration values. This effectively reduces the search space while allowing the agent to perform various tasks.

Table 5: Comparison of success rates across different scenarios and methods.

|  | HANSOME | DreamerV3-C | DreamerV2-C | Director-C |
|---|---|---|---|---|
| DenseTraffic | **88.17% ± 1.08%** | 40.66% ± 2.78% | 48.89% ± 4.44% | 66.67% ± 6.67% |
| Roundabout | **89.46% ± 3.10%** | 84.52% ± 2.38% | 88.89% ± 3.14% | N/A |
| LeftTurn | **85.19% ± 4.14%** | 80.16% ± 3.46% | 64.52% ± 3.65% | N/A |
| RightTurn | **94.27% ± 0.63%** | 90.42% ± 0.61% | 72.49% ± 3.85% | N/A |

**Action Space.** The higher-level one-hot intention and the lower-level one-hot control are then concatenated to form a two-hot action. We use two-hot action instead of one-hot action for the two-level policy to mitigate the sparsity of the action space.

### D.3 COMMUNICATION SETTINGS

Given our action space discussed in Appendix D.2, the intention messages are formatted in one-hot encoding for each sender. When multiple agents send messages, the message space can grow exponentially with the number of agents communicating with the ego vehicle. Therefore, it is crucial to introduce a strategy to communicate with the most relevant agents. Determining "whom to communicate with" in multi-agent environments is a highly non-trivial problem (Zhu et al., 2022). A common approach is to select nearby agents (Yun et al., 2021). In our work, we adopt this strategy by selecting the three nearest vehicles for all tasks. This approach is intuitive in human driving scenarios and performs reasonably well in our experiments. Investigating more complex communication protocols can be explored in future research. For different higher-level intentions, the ego agent may be interested in agents from different directions. For example, when the intention is to change lanes, the ego vehicle primarily focuses on the vehicle ahead or the nearest vehicles in neighboring lanes.

### D.4 OVERALL PERFORMANCE

In addition to Section 4, we present a comprehensive performance comparison between HANSOME and baseline models across all tasks in Table 5. This includes an additional challenging `Roundabout` scenario, which features aggressive and dense traffic. We also report baseline performance on `LeftTurn` and `RightTurn`, complementing the ablation study of HANSOME on these tasks.

We follow the same task and model configurations, as well as hyper-parameters, detailed in Appendix E and Appendix F. Baseline models adhere to their original implementations, sharing hyper-parameters for common components, and no additional hyper-parameter tuning was performed for any models on these tasks. Director does not actively explore during training and fails to complete the task within the same 600k training step budget, during which other models have already acquired the necessary skills. Due to its lower sample efficiency and the significantly longer time it requires to converge on these tasks, we mark its results as N/A.

## E    TASK CONFIGURATIONS

We use the same CARLA simulation and task settings across all the baselines.

Table 6 shows the configurations of CARLA simulation and our designed tasks, including a generic reward function that applies to all the tasks, and task-specific configurations such as traffic flow density. Note that the route following rewards are used across different baselines and HANSOME, while HANSOME reward is degraded when the lower-level is deviating from the higher-level policy's planned intentions.

In `LeftTurn` and `RightTurn`, all the background vehicles are aggressive autopilots in CARLA; the ego agent is at the crossing and must turn left or right, the higher-level policy does not take effect in this case since the turn route is enforced; the ego agent has to identify the optimal timing to merge in the traffic flow. In `DenseTraffic`, we use the CARLA map *Town04*, spawn and manage 300 background vehicles using *TrafficManager*.

| Name | Value |
|---|---|
| **Simulation** | |
| FPS | 0.1s |
| BEV size | $128 \times 128$ |
| Desired speed | 4 m/s |
| Maximum episode length | 1000 |
| **Action Spaces** | |
| Distribution | Two-hot encoding |
| Acceleration space | $0, \pm 2$ |
| Steering space | $0, \pm 0.2, \pm 0.6$ |
| Intention Space | go straight, left, right |
| **Reward Scales** | |
| Reaching waypoint | 2.0 |
| Parallel speed | 0.5 |
| Perpendicular speed | $-1.0$ |
| Collision | $-30$ |
| Deviation from waypoints | $-3.0$ |
| Deviation from intentions | 2.0 |
| Invalid intention | $-5.0$ |
| Reaching destination | 50.0 |
| **DenseTraffic** | |
| Background vehicle number | 300 |
| **LeftTurn** | |
| Distance between cars in traffic flows | 6m to 8m |
| **RightTurn** | |
| Distance between cars in traffic flows | 6m to 8m |
| **ObstacleBypass** | |
| Ego's distance to obstacle | 40m |

Table 6: Task configurations.

## F  MODEL CONFIGURATION

We use the same MLP and CNN sizes for HANSOME and all baseline models. To ensure fair comparison over baselines, we use "small" size DreamerV3 with original hyper-parameters from their paper (Hafner et al., 2023). The difference in network architecture lies in actor-critic - Dreamers are single actor-critic; Director uses two actor-critics and each contains two MLPs for actor and critic; HANSOME is a dual-head actor with a critic. The dual-head actor produces two-hot actions for each level to mitigate the sparsity of joint actions and allow WMs to imagine using both levels of actions.

Director's lower-level policy is driven by intrinsic rewards based on the cosine similarity of and the current observation image, and goal image planned by the higher-level every 16 steps. However, measuring goal completion through image similarity is not applicable to many of the tasks. Even though images can visualize a goal, they do not imply executable actions in a straightforward way, unlike HANSOME's intentions that enforce text semantics.

| Name | Value |
|---|---|
| **General** | |
| Batch size | 16 |
| Batch length | 64 |
| Replay buffer size | $10^6$ |
| Activation | SiLU |
| CNN layer | 32 |
| MLP layer | 2 |
| MLP hidden units | 512 |
| **World Model** | |
| Number of latents | 32 |
| Classes per latent | 32 |
| Memory units of RSSM | 512 |
| Reconstruction loss scale | 1.0 |
| Dynamics loss scale | 0.5 |
| Representation loss scale | 0.1 |
| Learning rate | $10^{-4}$ |
| Adam epsilon | $10^{-8}$ |
| **Actor Critic** | |
| Imagination horizon | 15 |
| Return lambda | 0.95 |
| Return normalization limit | 1 |
| Return normalization decay | 0.99 |
| Actor entropy scale | $3 \times 10^{-4}$ |
| Learning rate | $3 \times 10^{-5}$ |
| Adam epsilon | $10^{-5}$ |
| Gradient clipping | 100 |

Table 7: Parameters for HANSOME and Dreamers.

## G  MORE ABSTRACT OVERVIEW OF HANSOME

We provide another high-level illustration in Figure 13 to showcase the HANSOME agents' interactions through seamless integration of semantic communication and hierarchical planning. Visual information is shared to provide enriched BEVs in complex traffic scenarios. Agents fuse and leverage enriched BEVs and shared intentions (text instructions) from other agents to predict background agents' trajectories and thereby enhance safety. Aside from aiding other agents' planning, the intention can also aid lower-level policy by providing guidance towards the given destination.

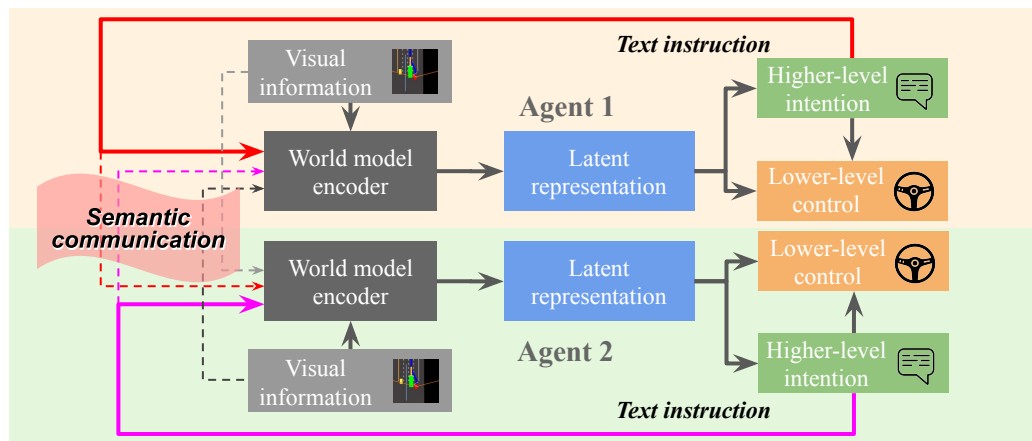

Figure 13: Agents communicate through two common "languages": high-level text instruction for sharing intention information and lower-level BEV semantics for sharing visual information.

