# OpenReview forum: "World-Model based Hierarchical Planning with Semantic Communications for Autonomous Driving"
_ICLR.cc/2025/Conference — Submitted to ICLR 2025_

### Official Review · Reviewer_Z1hx · 2024-11-04

**Soundness:** 2
**Presentation:** 2
**Contribution:** 2
**Rating:** 5
**Confidence:** 4

**Summary:**

This paper proposes, HANSOME, a WM-based hierarchical planning model for AV under the V2V setting (the intention of other vehicles can be accessed by the ego vehicle). The proposed model uses the Dreamer backbone to train the high-level policy (intent conditioned trajectory prediction) and the low-level policy (waypoints following) through an entropy-controlled adaptive scalarization approach. The paper is well-written and easy to follow.

**Strengths:**

The paper is well-written and easy to follow. The proposed approach is validated through multiple experiment configurations to demonstrate its effectiveness compared to baselines.

**Weaknesses:**

1. The motivation of the low-level policy. Line276-279 indicate that the intention of the high-level policy is rendered as the waypoints on the bev, and the low-level policy just tries to track the waypoints. This means that the low-level policy is equivalent to a PID controller and is independent of tasks. So what is the motivation to include in the learning problem? To justify this, I encourage the authors to demonstrate that why directly calling a tuned PID waypoint tracking algorithm or an independently learned tracker is not good compared to jointly learn a tracker and motion planner.

2. The need of a simulator. The loss needs to compute the tracking error (eq 1). Does that mean you still need a simulator during policy rollout? This is a bit strange, because the motivation for learning a world model is to learn a realistic simulator. I think this work does three things: 1) an intent conditioned traj prediction (the encoder-decoder model) model; 2) a waypoints tracker model; 3) a loss that trains these models together. I feel that 1 is not new, and  3) is not necessary.

3. I encourage the authors to validate the proposed approach in Waymax in the future, which provides more traffic scenarios (> 100k) compared to just using 4 scenarios in the current version.

**Questions:**

Please see the comments in the Weaknesses section.

---

> ### Author Response · Authors · 2024-11-23
> **Responses to Reviewer Z1hx**
>
> **Last Update: in response to the reviewer's comment on Dec 2nd**
>
> We thank the reviewer for reviewing our paper, and we address the raised concerns as follows.
>
> ## [W1] The reviewer inquired about the motivation to learn the lower-level policy instead of directly tuning a PID controller to track waypoints.
>
> We thank the reviewer for providing a different perspective, and we clarify that an end-to-end lower-level policy is favorable  and we summarize the reasons as follows:
>
> a) Decision Making on a Rich Context. The lower-level policy operates on rich latent representations encoding complex state information including traffic dynamics derived from image inputs, not just positions or speeds used by a PID controller. This allows HANSOME to learn adaptive behaviors for collision avoidance and speed modulation directly based on visual inputs with surrounding traffic.
> For example, the higher-level policy may plan the same “go straight” intention in different situations, but the surrounding traffic makes a huge difference in expected behaviors of the lower-level policy. The lower-level policy can decide to speed down to follow leading vehicle or keep its current speed when there is no leading vehicle. It goes beyond the capability of a traditional PID controller solely based on position and speed. In contrast, our lower-level policy can handle such cases since it plans based on richer information of the surrounding environment in an end-to-end manner. Our demos in supplementary materials (e.g., collision avoidance) also showcase such capabilities of the lower-level policy that behaves differently given different traffic situations.
>
> (b) Joint Optimization for Optimality. Joint training of high and low-level policies enables more coordinated behavior adaptation. The high-level policy can learn to generate waypoints that account for the low-level policy's capabilities, while the low-level policy simultaneously learns to better interpret and execute the high-level intentions. This tight coupling would be lost with an independently tuned tracking controller. Moreover, it has been widely demonstrated that end-to-end policies are highly promising [1,6,7,11,12]. HANSOME’s lower-level policy also does not involve physical parameter tuning like Ki, Kp, and Kd in PID controllers.
>
> ## [W2] The reviewer inquired why we need a simulator during policy rollout, and the reviewer suggested “the motivation for learning a world model is to learn a realistic simulator”.
>
> We clarify that during policy rollout, the tracking error from Eqn. 1 the reviewer referred to (typically referred to as reward) is directly predicted by the world model without the assistance of the simulator. Policy training occurs entirely in the world model's imagination (inference), not in simulation.
> We utilize the simulator to compute ground truth reward, but that is for the purpose of training the world model to predict the reward. Then, the agent learns its policy solely based on the learned dynamics and reward model. This aligns with the standard practice in world model works including DreamerV1-V3 [2][3][4] and MBRL works [5].
>
> ## [W3] Benchmark
>
> We greatly appreciate the reviewer's suggestion regarding Waymax benchmark. We would like to clarify our choice of CARLA as our primary testing environment for the following crucial considerations.
>
> First,  Waymax challenges have mainly focused on predictive tasks, while HANSOME focuses on closed-loop interactive scenarios for policy learning and evaluation.  CARLA has been widely used in the well-established works on  imitation learning or reinforcement learning to evaluate their policies on customized scenarios (like MILE[6], SEM2 [7], TransFuser [8], InterFuser [9], ISO-Dream [11]). Moreover, we note that the 100k scenarios in Waymax mentioned by the reviewer are logged records driven by IDM. Therefore, the Waymax is suitable for predictive tasks like predicting future trajectory, occupancy flow, or semantic segmentation [10].
>
>
> We hope the clarifications can address the reviewer’s concerns and we are willing to address any further questions.

---

> > ### Author Response · Authors · 2024-11-23
> > **References**
> >
> > [1] Chen, Li, et al. "End-to-end autonomous driving: Challenges and frontiers." IEEE Transactions on Pattern Analysis and Machine Intelligence (2024).
> >
> > [2] Hafner, Danijar, et al. "Dream to control: Learning behaviors by latent imagination." arXiv preprint arXiv:1912.01603 (2019).
> >
> > [3] Hafner, Danijar, et al. "Mastering atari with discrete world models." arXiv preprint arXiv:2010.02193 (2020).
> >
> > [4] Hafner, Danijar, et al. "Mastering diverse domains through world models." arXiv preprint arXiv:2301.04104 (2023).
> >
> > [5] Ha, David, and Jürgen Schmidhuber. "World models." arXiv preprint arXiv:1803.10122 (2018).
> >
> > [6] Hu, Anthony, et al. "Model-based imitation learning for urban driving." Advances in Neural Information Processing Systems 35 (2022): 20703-20716.
> >
> > [7] Gao, Z., Mu, Y., Chen, C., Duan, J., Luo, P., Lu, Y., & Li, S. E. (2024). Enhance sample efficiency and robustness of end-to-end urban autonomous driving via semantic masked world model. IEEE Transactions on Intelligent Transportation Systems.
> >
> > [8] Chitta, Kashyap, et al. "Transfuser: Imitation with transformer-based sensor fusion for autonomous driving." IEEE Transactions on Pattern Analysis and Machine Intelligence 45.11 (2022): 12878-12895.
> >
> > [9] Shao, Hao, et al. "Safety-enhanced autonomous driving using interpretable sensor fusion transformer." Conference on Robot Learning. PMLR, 2023.
> >
> > [10] Gulino, Cole, et al. "Waymax: An accelerated, data-driven simulator for large-scale autonomous driving research." Advances in Neural Information Processing Systems 36 (2024).
> >
> > [11] Pan, Minting, et al. "Iso-dream: Isolating and leveraging noncontrollable visual dynamics in world models." Advances in neural information processing systems 35 (2022): 23178-23191.
> >
> > [12] Li, Qifeng, et al. "Think2drive: Efficient reinforcement learning by thinking in latent world model for quasi-realistic autonomous driving (in carla-v2)." arXiv preprint arXiv:2402.16720 (2024).

---

> > > ### Comment · Reviewer_Z1hx · 2024-12-02
> > >
> > > I would like to thank the author's responses.
> > >
> > > For my first concern, the author's states that jointly learning the low-level policy could help high-level policy learn to generate waypoints that account for the low-level policy's capabilities. I am wondering if training these two tasks will lead to a suboptimal solution? Like the high-level policy knows that the low-level tracker is not good, so it gives up and never generates an optimal solution. The provided references do not provide any information on why co-learning path tracker is better than using a well-designed tracker, which is not hard and is the current industry standard.
> > > I would also like to point out that each vehicle is different, even if they are from the same brand. Thus, what's happening in reality is that many vehicles have different config files for their path tracker due to their differences in the vehicle system. Co-training the path-tracker in your planning algorithm is very hard if the goal is to deploy a fleet, which means some cars will need to have different models. I encourage the authors to state what's the benefit and what's the limitation brought by this design choice.
> > >
> > > Since the authors have addressed my second concern, I will change the score to 5.

---

> > > > ### Author Response · Authors · 2024-12-03
> > > > **Response to Reviewer Z1hx**
> > > >
> > > > We thank the reviewer for the insightful questions regarding optimality of co-learning and heterogeneous agent models, which are  exactly the subjects of proposed AdaSMO and ego-centric learning:
> > > >
> > > >
> > > > 1. **"Whether co-learning leads to suboptimal solutions?"**
> > > >    **"Like the high-level policy knows that the low-level tracker is not good, so it gives up and never generates an optimal solution."**
> > > >
> > > >
> > > >   AdaSMO addresses this co-learning challenge by adaptively adjusting the entropy of the higher-level policy. This approach is inspired by the human learning process, where humans first develop diverse basic skills before integrating them for complex tasks. Specifically:
> > > >    - **HANSOME uses the higher-level policy to guide lower-level policy learning**. When the lower-level policy is still underdeveloped, AdaSMO increases the entropy of the higher-level policy, enabling the lower-level policy to explore and learn diverse basic skills.
> > > >    - **As the lower-level policy improves**, AdaSMO reduces the entropy of the higher-level policy while simultaneously increasing task difficulty (e.g., transitioning from sparse to dense traffic or from fixed to dynamic higher-level goal horizons).
> > > >
> > > > The ablation in Section 4.4 demonstrates the effectiveness of AdaSMO to avoid learning suboptimal policy.
> > > >
> > > > Additionally, we clarify that HANSOME’s lower-level policy is not merely a tracker; it enables decision-making based on latent representations with rich contextual information.
> > > >
> > > >
> > > > 2. The provided references ([1, 6, 7, 8, 11, 12]) are surveys and approaches for end-to-end autonomous driving. These works demonstrate the potential of end-to-end methods. For example, works like [6, 7, 11, 12] use CARLA map routes as agent inputs, enabling their policies to track waypoints in an end-to-end manner rather than relying on a PID controller.
> > > >
> > > >
> > > > 3. **"Co-training the path-tracker in your planning algorithm is very hard if the goal is to deploy a fleet, which means some cars will need to have different models."**
> > > >
> > > >
> > > >    We agree with the reviewer’s point that real-world traffic systems consist of vehicles with varying models and configurations. Consequently,
> > > >    - **HANSOME does not assume uniform models across agents** (lines 81-90). We clarify that HANSOME agents do not co-train path-trackers across agents. Instead, we adopt ego-centric learning (Appendix B).
> > > >    - HANSOME agents are designed to interact with agents of different models and configurations. For example: In our experiments, HANSOME agents interact with other HANSOME agents, rule-based agents (via ego-centric learning), and even aggressive agents (e.g., LeftTurn, RightTurn, Roundabout) that mimic irrational driver behaviors. HANSOME also demonstrates compatibility with varying communication configurations, as shown in Table 3.
> > > >
> > > > ---
> > > >
> > > > We have revised the paper based on the reviewer's feedback to clarify the role of world models (lines 236-238) and to explain why HANSOME is capable of interacting with heterogeneous agents in real-world traffic systems (lines 81-90). **We hope that our responses, along with the revisions, address the remaining concerns of the reviewer. If so, we kindly request you to consider revising the score before the discussion period ends today.**

---

### Official Review · Reviewer_qDEw · 2024-11-04

**Soundness:** 3
**Presentation:** 3
**Contribution:** 3
**Rating:** 5
**Confidence:** 3

**Summary:**

This work introduces a method named HANSOME for autonomous driving that combines hierarchical planning with semantic communication in a Dreamer V3 framework. It adopts a hierarchical reinforcement learning structure where the higher-level policy sets semantic intentions, and the lower-level policy determines control actions based on these intentions. An adaptive scalarization method AdaSMO that dynamically balances multi-objective optimization between the hierarchical levels is also proposed.

**Strengths:**

- Easy to follow. Clear writing.
- The idea of incorporating language into the Dreamer V3 framework is straightforward and can lead to better performance.
- The illustration of intention and sharing across multiple agents are interesting.
- The AdaSMO method dynamically adjusts the focus between high-level and low-level objectives, allowing for more stable training and performance optimization across hierarchical levels.
- Better performance compared to baselines and very good visualizations provided.

**Weaknesses:**

- The authors mainly use the toy testing scenario, only considering left turn, right turn, and merging. These simulated scenarios can be easily to be Could the author try complex scenarios or simulation environments?
- The model’s performance highly relied on the quality of semantic intentions, which may not always be accurate or available in real-world settings.
- The work is more like integrating language into the Dreamer framework,  lacking novelties.
- How does the model understand language or traffic rules? Did the author provide a text prompt of traffic rules or involve large language models?
- Could the framework handle unpredictable human driver behavior in mixed traffic environments?
- Could this framework framework be generalized to other complex, multi-agent environments beyond autonomous driving?
- The proposed AdaSMO looks like human-crafted parameter tuning, which is not considered adaptive.

**Questions:**

Please see the weaknesses part. Thanks.

---

> ### Author Response · Authors · 2024-11-23
> **Responses to Reviewer qDEw (Part 1)**
>
> We thank the reviewer for the careful review and feedback. We will provide more clarifications in response to the reviewer’s concerns.
>
> ## [W1] Testing Scenarios
>
> Thank you for the suggestion. We add a new challenging scenario “Roundabout” with dense and aggressive traffic on top of the scenarios we already have (DenseTraffic, LeftTurn and RightTurn, ObstacleBypass). We keep the same model configurations and hyper-parameters, and used 3 random seeds to compute the confidence interval. Director failed to learn a reasonable policy with similar training budgets as other models around 600k steps, therefore we marked it as N/A. Overall, HANSOME shows optimal performance on all tasks.
>
> | **Scenario**       | **HANSOME**           | **DreamerV3-C**       | **DreamerV2-C**       | **Director-C**      |
> |---------------------|-----------------------|-----------------------|-----------------------|---------------------|
> | **Dense Traffic**   | 88.17% ± 1.08%       | 40.66% ± 2.78%        | 48.89% ± 4.44%        | 66.67% ± 6.67%      |
> | **Roundabout**      | 89.46% ± 3.10%       | 84.52% ± 2.38%        | 88.89% ± 3.14%        | N/A                 |
> | **Left Turn**       | 85.19% ± 4.14%       | 80.16% ± 3.46%        | 64.52% ± 3.65%        | N/A                 |
> | **Right Turn**      | 94.27% ± 0.63%       | 90.42% ± 0.61%        | 72.49% ± 3.85%        | N/A                 |
>
>
> “-C” indicates the baseline models benefits from enhanced observability through semantic communications for fair comparison.
>
> We clarify that it is non-trivial to design testing scenarios involving RL based online learning approached with communications.
>
> We discussed the considerations of task design from line 868-873 and 910-925, and discussed how prior works customized their tasks for evaluation. For example, some prior world model based RL works test the agent to drive to maximize total rewards within 1000 steps [11, 13] in CARLA town03 with 20 or 100 vehicles. HANSOME is online RL learning and our tasks are more challenging in the sense that traffic is even denser (300 vehicles in DenseTraffic) and the task goal is not to drive as much as possible but to complete a driving skill. Collision avoidance is way more challenging at intersections since the traffic flow is set to be “aggressive” and they can ignore the ego vehicle like irrational human drivers. The ego agent has to understand other agents’ intentions and predict whether they will interfere with the ego agent to find a proper time to proceed. Furthermore, for our work that features multi-agent communications, it is also necessary for us to design and develop tasks that rigorously evaluate the impact of semantic communications. But these tasks are not readily available, and this is why we have meticulously designed such challenging tasks with dense traffic or challenging objective in a highly interactive environment CARLA. The ego vehicle has the find proper timing to merge lanes, or take turns, or flexibly bypass obstacles ahead, avoid collision, and complete tasks.
>
> ## [W2] Availbility of Intentions in Real-World Settings.
>
> In response to the reviewer's concern, we add experiments to show that HANSOME are robust to environments where semantic intentions are unavailable.
>
> | **Communications**         | **LeftTurn**       | **RightTurn**       |
> |-----------------------------------|--------------------|---------------------|
> | w/ visual only     | 82.21% ± 6.94%     | 91.62% ± 3.04% |
> | w/ visual + intention (HANSOME)            | 85.19% ± 4.14%     | 94.27% ± 0.63%      |
>
> ## [W4]  How does the model understand language or traffic rules
>
> HANSOME agents understand intentions through a cross-modal encoder-decoder. The decoder tries to predict other vehicles’ future trajectories from latent representations. This can be seen as an auxiliary task to force the latent representations to learn what is means by each intention, while the agent can still learn purely from visual inputs without additional text prompts nor prior knowledge.
> We did not use LLMs nor text prompts. HANSOME is a reinforcement learning based framework like [2, 3, 6], where the expected agent behaviors are defined by reward functions. We do not explicitly provide traffic rules to the agent. Instead, we include reward terms such as these penalizing out-of-lane or collisions.

---

> > ### Author Response · Authors · 2024-11-23
> > **Responses to Reviewer qDEw (Part 2)**
> >
> > ## [W3] Novelty Clarification
> >
> > We would like to clarify our novelty here. We are not simply integrating language modality to Dreamer. We need to deal with these challenges to make the higher-level planner support language:
> >
> > 1.	How to concurrently learn the higher-level and lower-level policy and address non-stationarity when two policies are co-evolving?
> > 2.	How to align higher-level semantics and lower-level controls? How to interpret semantic messages from other agents?
> >
> >
> > **We propose new designs that address the above challenges.**
> >
> > 1.	We propose AdaSMO, to mitigate non-stationarity in hierarchical policy learning, a long-standing challenge [1] in RL that was used to be mitigated through re-labelling techniques – while HANSOME does not need extra re-labelling process, and we directly manipulation policy exploration to fix the issue.
> > 2.	We propose semantic alignment for the ego agent – translating text-based intentions to visual information on BEVs. Moreover, to interpret other agents’ intentions, we leverage a cross-modal encoder-decoder, to translate shared intentions to visual information on BEVs.
> >
> >
> > Moreover, we are considering agents sharing their intentions in a multi-agent system, which is distinguished from prior works in closed-loop E2E autonomous driving [2, 3, 4, 5, 6] that do not incorporate vehicle communications.
> >
> > ## [W5] Unpredictable human driver behavior
> >
> > Thank you for pointing this out. We clarify that in LeftTurn, RightTurn, and the new task Roundabout, the traffic flows are set to be aggressive autopilots, and they will not actively avoid collisions, similar to unpredictable or irrational human drivers.  HANSOME agents are robust to these as shown in the response to [W1] or Table 3 in the paper.
> >
> > ## [W6] Other Application Domains
> >
> > HANSOME’s generalization to other domains beyond autonomous driving (AD). It can be highly beneficial to extend HANSOME to other task domains like cooperative robots for future works, though these domains are not the focus on this work, since developing world models or communicative agents for AD is highly non-trivial [7]. Many prior world model or RL based works focus on developing a specialized approach just for the AD domain [2, 3, 4, 5, 6].
> >
> > In response to the reviewer’s question, we would love to discuss HANSOME’s potential to other task domains given the several designs of HANSOME that are general and can potentially facilitate world model learning in other domains.
> >
> > 1)	**AdaSMO** can be generalized to different task domains, because the exploration is inherent to online RL policy learning [8]. AdaSMO does not require re-labelling which updates collect trajectories with a defined goal state, unlike prior hierarchical RL works [1] using hindsight replay.
> > 2)	**Ego centric learning** discussed in Appendix B makes learning in multi-agent systems with communications more practical. HANSOME shows generalizability to multi-agent testing where multiple HANSOME agents can interact even HANSOME is trained in a distributed ego-centric manner. The idea here can be leveraged by other multi-agent tasks.
> > There are also technical aspects in HANSOME specifically designed to address challenges specific to AD. For example, the cross-modal encoder-decoder is a specialized designed for AD to effectively fuse driving intentions with BEVs and decode the latent representation to BEVs with visual waypoints. The cross-modal learning can be thought of as an auxiliary (not mandatory) task that enhances message understanding in AD. Therefore, it is also possible to adapt to other task domains by removing auxiliary tasks or adopting different message fusion module based on the specific task requirements.
> >
> > ## [W7] Why AdaSMO is Adaptive
> >
> > AdaSMO is adaptive since the exploration of hierarchical policy is dynamically adjusted according to the skill level of policy - similar to human learning to integrate different basic skills after all individual skills are proficient. Therefore, adaptive scalarization parameters vary across different tasks, because the definition of "skill level of policy" (rewards in RL) varies. For example, in AD, it can be the proficiency/rewards of steering or pedel control skills, and this is necessary to quatify the quality of lower-level policy. Similar to the learning rate decay that differs from dataset to dataset for neural networks training, once the task-specific parameters of AdaSMO are determined, it will adaptively adjust the exploration along the learning process.
> >
> > We hope the clarifications can address the reviewer’s concerns and we are willing to address any further questions.

---

> > > ### Author Response · Authors · 2024-11-23
> > > **References**
> > >
> > > [1] Pateria, Shubham, et al. "Hierarchical reinforcement learning: A comprehensive survey." ACM Computing Surveys (CSUR) 54.5 (2021): 1-35.
> > >
> > > [2] Hu, Anthony, et al. "Model-based imitation learning for urban driving." Advances in Neural Information Processing Systems 35 (2022): 20703-20716.
> > >
> > > [3] Gao, Z., Mu, Y., Chen, C., Duan, J., Luo, P., Lu, Y., & Li, S. E. (2024). Enhance sample efficiency and robustness of end-to-end urban autonomous driving via semantic masked world model. IEEE Transactions on Intelligent Transportation Systems.
> > >
> > > [4] Chitta, Kashyap, et al. "Transfuser: Imitation with transformer-based sensor fusion for autonomous driving." IEEE Transactions on Pattern Analysis and Machine Intelligence 45.11 (2022): 12878-12895.
> > >
> > > [5] Shao, Hao, et al. "Safety-enhanced autonomous driving using interpretable sensor fusion transformer." Conference on Robot Learning. PMLR, 2023.
> > >
> > > [6] Li, Qifeng, et al. "Think2drive: Efficient reinforcement learning by thinking in latent world model for quasi-realistic autonomous driving (in carla-v2)." arXiv preprint arXiv:2402.16720 (2024).
> > >
> > > [7] Chen, Li, et al. "End-to-end autonomous driving: Challenges and frontiers." IEEE Transactions on Pattern Analysis and Machine Intelligence (2024).
> > >
> > > [8] Ladosz, Pawel, et al. "Exploration in deep reinforcement learning: A survey." Information Fusion 85 (2022): 1-22.

---

> > > > ### Comment · Reviewer_qDEw · 2024-11-24
> > > >
> > > > I appreciate the author's detailed response. The response solved most of my questions, I would like to keep my original rating.

---

> ### Author Response · Authors · 2024-12-03
> **Response to Reviewer qDEw**
>
> We thank the reviewer qDEw for their thoughtful review and questions, which have greatly helped us improve our paper. We're glad the most of the reviewer's concerns were addressed! Based on the reviewer’s feedback, we have made the following revisions:
>
> - **Handling Unpredictable Driving Behaviors**: We show that the agent can handle unpredictable driving behaviors by introducing aggressive background vehicles in our urban driving tasks. This is addressed from lines 405, 474, and Appendix D.4.
> - **Practicality of Intentions**: We discuss the impact of the availability of intentions as a practical issue in real-world settings (line 322). We demonstrate that HANSOME does not assume full availability of intentions while still achieving reasonable performance (Table 3).
> - **Adaptivity of AdaSMO**: We provide further discussion on why AdaSMO is adaptive and how it is inspired by adaptive learning process of human (line 1144, line 356, line 367, and Appendix D.1).
> - **Comparisons with More Literatures**: We included more comparisons to HRL, MARL in Related Works and Appendix A, highlighting the novelty of HANSOME compared to these fields.
>
> *We are happy to address any additional questions on the final day of the discussion period. We hope that our responses and revisions have resolved the remaining concerns. If so, we kindly request the reviewer to consider updating the score before the discussion period ends today.**

---

### Official Review · Reviewer_cXkR · 2024-11-04

**Soundness:** 3
**Presentation:** 3
**Contribution:** 2
**Rating:** 6
**Confidence:** 4

**Summary:**

This paper proposes a world-model-based planner for autonomous driving, named HANSOME. HANSOME first predicts the high-level intention based on the hidden state and observation. Then it predicts both the low-level control signal and waypoints on BEV maps conditioned on the intention. The world model or latent encoding part takes BEV and intentions of other vehicles (waypoints under BEV) as input and reconstructs future trajectories of neighbors. HANSOME designs reward by combining intention generation and waypoint following in low-level policies. To learn the two levels of planning in HANSOME, the authors apply various stages of scalarization to control the entropy of high-level policy outputs and balance the ultimate optimization weight. HANSOME is tested on four scenarios on CARLA and outperforms Dreamer-based baselines. Ablations are also conducted to validate the effectiveness of the modules in HANSOME.

**Strengths:**

1. The paper is easy to follow and understand overall. Discussions on related works are relatively sufficient for understanding their differences. Even end-to-end driving and hierarchical planner in embodied AI are discussed in the appendix as well, which is very good and comprehensive.
1. The evaluation is adequate overall. Essential ablations have been conducted for different designs of HANSOME. I am glad to see the multi-agent learning experiments in the appendix, though the setting is relatively simple. As the paper highlights semantic information sharing, multi-agent or V2X settings should be validated to highlight its effectiveness.
1. The source code is provided and detailed parameters are listed in the appendix for reproduction.

**Weaknesses:**

- Multiple designs in HANSOME have been validated for their effectiveness, or widely adopted, in other areas. The semantic information sharing is widely used in V2X and multi-agent approaches, and the authors have cited some of these works. Hierarchical planning is also a common way in the industry, and a lot of language-related driving papers such as DriveVLM [1] and DriveVLM [2]. Predicting control signals and waypoints at the same time is used in previous works like TCP [3]. The world model predicts other vehicles future motion and does not feature very novel designs from my viewpoint.
- There seem a lot of heuristic settings or designs in HANSOME. Therefore, though HANSOME achieves much better performance compared to DreamerV2&V3, I am wondering if the effectiveness is carefully tuned and worrying about its broader impacts. For example, HANSOME uses a heuristic method to select agents for information exchange; the adaptive scalarization constant S is heuristically adjusted with various stages; the reward designs.
- Though the authors claim that HANSOME agents can generate intentions by themselves while other works like MILE, SEM2, and Think2Drive need pre-determined routes for guidance, I think this advantage is because current evaluations are solely conducted in a very small scenario, like LeftTurn and RightTurn. I also get why the authors mention the comparisons with route planning in Lines 89-97. However, in this work, HANSOME does not include route planning in its structure which limits its long-term planning ability.
- Based on the previous point, I believe the current benchmarks, four specifically collected scenarios, are relatively simple. Maybe it is a common way for world model-based methods, but standard evaluation setups like Town05Long or leaderboard v1&v2 should be much more convincing.
- The method is not strictly end-to-end planning as its inputs are BEV rendering images, not raw sensory inputs.
- Minors.
  - I do not see waypoints or destination directions of other vehicles in BEV in Fig. 3.
  - Typos. Line 151, ``generated by the''

[1] Tian, Xiaoyu, et al. "Drivevlm: The convergence of autonomous driving and large vision-language models." arXiv preprint arXiv:2402.12289 (2024).
[2] Sima, Chonghao, et al. "Drivelm: Driving with graph visual question answering." arXiv preprint arXiv:2312.14150 (2023).
[3] Wu, Penghao, et al. "Trajectory-guided control prediction for end-to-end autonomous driving: A simple yet strong baseline." Advances in Neural Information Processing Systems 35 (2022): 6119-6132.

**Questions:**

- Could you provide the whole results under LeftTurn and RightTurn, including the results of other baselines? These results should show the effectiveness of hierachical planning and AdaSMO.

---

> ### Author Response · Authors · 2024-11-23
> **Responses to Reviewer xCkR (Part 1)**
>
> We thank the reviewer for the careful review and insightful feedback. We address the raised concerns as follows.
>
> ## [W1] More Comparison with Related Works
>
> We thank the reviewer for sharing the concern. We will provide detailed comparison against the research fields suggested by the reviewer and will revise the paper.
>
> ### 1) Comparison with LLM-Based Methods (e.g., DriveLM [6] and DriveVLM [7])
>
> LLM based methods such as DriveLM [6] and DriveVLM [7] are open-loop methods. The objective involves optimizing NLP-based metrics, such as comparing scene descriptions/analysis with ground-truth annotations (driving captions, visual Q&A) using GPT. The “hierarchical planning” in these works does not involve multiple policies – it proposes action descriptions and converts descriptions to waypoint tokens. Waypoint tokens are obtained from trajectory statistics in training data since having LLMs produce numerical results is challenging [6]. To evaluate LLM action proposals, they compare the converted trajectories with ground-truth trajectories. We can see the planning is achieved through LLM proposals that are mapped to waypoints, and they do not have an actual controller to track waypoints or evaluate the planner since it is open-loop. This is different from HANSOME that focuses on developing a closed-loop policy and evaluating policy performance in simulation. The lower-level policy follows accurate waypoints proposed by the higher-level planner. HANSOME higher-level policy produces semantic intentions without prior knowledge from LLMs. Both levels are learned from scratch within world models’ imagination without strong priors from LLMs. HANSOME models are more lightweight (30M parameters, significantly fewer than LLMs) and therefore more practical for real-time inference.
>
> ### 2) Comparison with Other Hierarchical Planning Works
>
> it is challenging and highly non-trivial to address non-stationarity during hierarchical RL training because multiple RL policies simultaneously change [9]. Imagine a higher-level policy plans intentions which cannot be achieved by the lower-level policy learning from scratch, and therefore the higher-level only observes short paths and cannot learn complex path planning skills. A vanilla two-policy training will yield poor performance as shown in Fig. 5. Conventional approaches of hierarchical RL leverage hindsight replay and re-labelling techniques to address non-stationarity. One of the contributions of HANSOME is AdaSMO that does not need any extra re-labelling process. AdaSMO is inspired by human learning, where humans learn each basic skill proficiently before integrating all skills for more complex tasks. AdaSMO adaptively adjusts higher-level policy entropy to scalarize the two-level learning. As shown in our ablation of AdaSMO and Fig. 5, we provided a novel perspective in solving hierarchical RL, and we successfully applied this hierarchical RL method in challenging autonomous driving tasks.
>
> ### 3) Comparison with Multi-Agent RL and V2X Research
>
> Although multi-agent RL or V2X communities have studied information sharing [1][11], it is challenging to apply information sharing to highly realistic simulation environments for closed-loop evaluation in addition to multi-agent communications. MARL typically suffers from high-dimensionality and non-stationarity [1], the common testbeds can be grid worlds like predator prey. We simplify the problem and leverage ego-centric learning for autonomous driving to make learning with communications tractable in multi-agent systems as discussed in Appendix B. Moreover, our closed-loop evaluation emphasizes more on policy performance. V2X research such as cooperative perception [11], can focus on, e.g., prediction errors of occupancy flow, that does not require policy rollouts. In summary, HANSOME aims at initiating the first step towards practical world model learning for autonomous driving with communications in multi-agent systems. The distinct challenges of closed-loop evaluation motivate us to pursue practical and novel solutions like semantic communications, ego-centric learning, and AdaSMO to mitigate issues of high-dimensionality and non-stationarity in multi-agent systems.
>
> ### 4) Comparison with TCP [8] and Works that Predict Vehicle Motions
>
> TCP is based on supervised imitation learning, while HANSOME is online RL based on world models. HANSOME needs to cope with non-stationarity during hierarchical RL learning. Though HANSOME supports two-level planning like TCP, and support predicting background vehicle future trajectories, it is neither the purpose of this work. The contribution of HANSOME is the seamless integration between the two – a higher-level policy that not only guides its own lower-level but also can generate messages to enhance others’ future prediction and planning.

---

> > ### Author Response · Authors · 2024-11-23
> > **Responses to Reviewer xCkR (Part 2)**
> >
> > ## [W2] Regarding "heuristic settings" in HANSOME, and why HANSOME does not need to be “carefully tuned” in our experiments
> > 1) **Selection of nearby agents for information exchange.** It is non-trivial to determine whom to communicate with in multi-agent RL [1]. It'll be an interesting extension for future works, but studying the communication protocol isn't the primary focus of our work, and selecting nearby agents is intuitive for normal driving and performance is reasonable.
> > 2) **Adaptive scalarization constant S.** Like the learning rate decay can differ across different datasets and tasks for training, the change of scalarization parameter S (rate decay in exploration) in AdaSMO also depends on the tasks, such as how difficulty is defined for driving tasks – it can be traffic density, or higher-level plan horizon, and this is the part where it is heuristic. However, just like that neural networks can be trained with different learning rates and decay to ultimately yield similar performance while there can be an optimal choice - as Figure 12 shows, AdaSMO still yields a good policy with different reward thresholds. In other words, AdaSMO can be robust to heuristic settings, and finding an optimal S is like tuning other hyper-parameters in training neural networks, but the reward thresholds are more intuitive – e.g., empirically, we found that it makes training more efficient to terminal lower-level skill warm-up at the point where the average rewards indicate all higher-level intentions can be proficiently completed.
> > 3) **Reward designs.** Reward engineering has been the main challenge in reinforcement learning. Extensive RL-based works for autonomous driving have studied various reward designs for driving tasks [3, 4, 5]. These reward functions are designed “heuristically” in some ways (like whether to penalize steering, whether to reward smoothness, how to reward positions) to reflect task-specific characteristics and expected behaviors of RL policies.
> > 4) We use the same the model configurations and hyper-parameters for all the baselines and HANSOME as discussed in Appendix F, except for model-specific components. For Director, we follow the original paper’s higher-level planner settings, and Director only supports a fixed time horizon for higher-level planner; for HANSOME, the scalarization constants are task-specific and are robust to different values as discussed in point (2) above. We also included our codes and all the model configurations in the supplementary materials for review.
> >
> > ## [W3] Why not provide route planning?
> >
> > As suggested by the reviewer, lines 89-97 mentioned route planning by Google Maps. In daily driving, there are three levels of planning 1) navigation maps provide route planning (e.g., exit highway after two miles), 2) human follow the navigation instructions and plan over a shorter horizon like merging lanes, which is the higher-level planner in HANSOME 3) plan concrete controls to achieve goals like merging lanes. It is not our focus to provide route planning at the first level, since it requires prior knowledge of map topology, and can be done by Google Maps. Our higher-level planner is at the second level and our lower-level policy is the third level. The planners of MILE, SEM2, or Think2Drive only has the third level. We will make this point clearer in the revised version.
> >
> > ## [W5] Using BEV is not strictly End-to-End Planning
> >
> > For RL based methods, it is a common practice to use BEV inputs [4]. The computer vision community also studies how to transfer sensory inputs to BEVs in an end-to-end manner [10], which can be integrated as an addition. Our work for now focuses on the planning with communications.
> >
> > ## [W6M1] Why there are no waypoints in the Fig. 3 BEV?
> >
> > This is expected, and we will provide more clarification for Fig. 3: waypoints of other vehicles only show up as the decoder outputs in the cross-modal encoder-decoder component. This is because the input BEVs do not have rendered waypoints, and other vehicles are sharing their intentions, and the ego vehicle learns to translate their shared intentions onto BEVs. Therefore, it is called cross-modal decoding. The decoder is not used during testing time, it can be thought of as an auxiliary task to help agents learn the semantics of shared intentions.
> >
> > ## [W6M2] Thank you for pointing out the typo. We will fix in the revision!

---

> ### Author Response · Authors · 2024-11-23
> **Responses to Reviewer xCkR (Part 3)**
>
> ## [W4] Benchmark Selection
>
> We thank the reviewer for the suggestion. HANSOME requires interactive and communicative scenarios to rigorously evaluate the benefits of semantic communications, while there are not such benchmarks readily available for closed-loop autonomous driving in highly realistic simulators like CARLA.
>
> Although it might take considerable effort to incorporate semantic communications to benchmarks like Leaderboard and address many potential challenges with online RL learning in multi-agent systems, we believe extending HANSOME to these even more complicated and challenging scenarios is a very meaningful step that we will pursue in future works.
>
> In recent studies, imitation learning approaches conduct more experiments on Leaderboard by having expert demonstrations [13]. We discussed in Appendix A that given the challenge to learn RL policies in online environemnts, previous world model based RL research like SEM2 [3] and Iso-Dream[12] tend to test on simpler driving cases with less traffic or easier tasks (20 or 100 vehicles in Town03). Think2Drive [10] (Feb 2024) is so far the only world model RL approach that trains DreamerV3 on Leaderboard scenarios, while it is not communicative and it uses BEV inputs with pre-determined routes. Given the challenge of both online RL learning and communicative learning in multi-agent systems, we will pursue this significant step for our future extensions, and we appreciate the reviewer's suggestion. Our tasks are challenging since they involve several driving skills, dense and aggressive background traffic. For example, we have 300 vehicles in DenseTraffic to test the agent's collision avoidance; in LeftTurn, RightTurn, and Roundabout, the dense traffic flows are set to be aggressive and they do not actively avoid collision (like irrational human drivers); ObstacleBypass is spefically designed to test the flexiblity of hierarchical planning.
>
> In response to the reviewer’s suggestion, we have added another challenging scenario Roundabout with dense traffic. We follow the same model settings/hyper-parameters discussed in Appendix F where baselines follow their original implementations and share hyper-parameters for common components. We did not do hyper-parameter tuning for any models like on other tasks. Director is inactive exploring during training and it fails to complete the task with the same 600k training step budget when other models already learns the skill. It is not as sample efficient as other models and might take much longer time than other models to converge, and therefore we mark the results as N/A.
>
> We will include the table in the response to Q1 below.
>
>
> ## [Q1] Baseline Performance on LeftTurn and RightTurn
>
> Thank you for the suggestion to further justify the effectiveness of hierarchical planning and AdaSMO. We added the experimented suggested by the reviewer and the table shows success rates of HANSOME and baseline models on LeftTurn, RightTurn, and a new task Roundabout. HANSOME shows optimal performance on all the tasks.
>
> | **Scenario**       | **HANSOME**           | **DreamerV3-C**       | **DreamerV2-C**       | **Director-C**      |
> |---------------------|-----------------------|-----------------------|-----------------------|---------------------|
> | **Dense Traffic**   | 88.17% ± 1.08%       | 40.66% ± 2.78%        | 48.89% ± 4.44%        | 66.67% ± 6.67%      |
> | **Roundabout**      | 89.46% ± 3.10%       | 84.52% ± 2.38%        | 88.89% ± 3.14%        | N/A                 |
> | **Left Turn**       | 85.19% ± 4.14%       | 80.16% ± 3.46%        | 64.52% ± 3.65%        | N/A                 |
> | **Right Turn**      | 94.27% ± 0.63%       | 90.42% ± 0.61%        | 72.49% ± 3.85%        | N/A                 |

---

> > ### Author Response · Authors · 2024-11-23
> > **References**
> >
> > We hope the clarifications can address the reviewer’s concerns and we are willing to address any further questions.
> >
> > ---
> > [1] Zhu, Changxi, Mehdi Dastani, and Shihan Wang. "A survey of multi-agent reinforcement learning with communication." arXiv preprint arXiv:2203.08975 1 (2022).,
> >
> > [2] Dewey, Daniel. "Reinforcement learning and the reward engineering principle." 2014 AAAI Spring Symposium Series. 2014.
> >
> > [3] Gao, Z., Mu, Y., Chen, C., Duan, J., Luo, P., Lu, Y., & Li, S. E. (2024). Enhance sample efficiency and robustness of end-to-end urban autonomous driving via semantic masked world model. IEEE Transactions on Intelligent Transportation Systems.
> >
> > [4] Li, Qifeng, et al. "Think2drive: Efficient reinforcement learning by thinking in latent world model for quasi-realistic autonomous driving (in carla-v2)." arXiv preprint arXiv:2402.16720 (2024).
> >
> > [5] Zhang, Zhejun, et al. "End-to-end urban driving by imitating a reinforcement learning coach." Proceedings of the IEEE/CVF international conference on computer vision. 2021.
> >
> > [6] Sima, Chonghao, et al. "Drivelm: Driving with graph visual question answering." arXiv preprint arXiv:2312.14150 (2023).
> >
> > [7] Tian, Xiaoyu, et al. "Drivevlm: The convergence of autonomous driving and large vision-language models." arXiv preprint arXiv:2402.12289 (2024).
> >
> > [8] Wu, Penghao, et al. "Trajectory-guided control prediction for end-to-end autonomous driving: A simple yet strong baseline." Advances in Neural Information Processing Systems 35 (2022): 6119-6132.
> >
> > [9] Pateria, Shubham, et al. "Hierarchical reinforcement learning: A comprehensive survey." ACM Computing Surveys (CSUR) 54.5 (2021): 1-35.
> >
> > [10] Li, Zhiqi, et al. "Bevformer: Learning bird’s-eye-view representation from multi-camera images via spatiotemporal transformers." European conference on computer vision. Cham: Springer Nature Switzerland, 2022.
> >
> > [11] Cui, Guangzhen, et al. "Cooperative perception technology of autonomous driving in the internet of vehicles environment: A review." Sensors 22.15 (2022): 5535.
> >
> > [12] Pan, Minting, et al. "Iso-dream: Isolating and leveraging noncontrollable visual dynamics in world models." Advances in neural information processing systems 35 (2022): 23178-23191.
> >
> > [13] Shao, Hao, et al. "Safety-enhanced autonomous driving using interpretable sensor fusion transformer." Conference on Robot Learning. PMLR, 2023.

---

> > > ### Comment · Reviewer_cXkR · 2024-11-24
> > > **Response to Authors' Rebuttal**
> > >
> > > Thanks for the authors' detailed reply. After reading other reviewers' comments and authors' replies, I decide to maintain my score.
> > >
> > > Basically, I share similar concerns with other reviewers about (1) the experimental setting; (2) novelty.
> > > (1) Note that Think2Drive mentioned in the authors' rebuttal was initially publicly shared in Feb 2024. As the authors mentioned in other replies, communication is not mandatory and the method can still achieve comparable performance with perturbations. Therefore, I think it would be better to position this paper without semantic communication as a core contribution. Otherwise, I think it is necessary to compare with certain baselines from V2X works.
> > > (2) Thanks for the detailed comparisons with other works. I find some comparisons (e.g. comparisons to V2X perception research) are very nuanced, which may also be considered minor or adaptation to the specific application.
> > > I hold my initial ideas for some other points that the authors claim are not the focus, such as BEV vs E2E and more complex scenarios or benchmarks with route planning.
> > >
> > > I maintain my score as 6 (marginally above the acceptance threshold) mainly because I think the experiments presented are sound, the idea is straightforward and the writing is clear.

---

> ### Author Response · Authors · 2024-11-28
> **Response to Reviewer xCkR**
>
> We thank the reviewer for carefully reading our responses and appreciate the reviewer’s constructive feedback. Based on your feedback, we have revised the paper to improve the clarity (the updates are highlighted in **blue fonts**).
>
> The revision includes the new experiments (Table 5) suggested by the reviewer, dicussions according to the reviewer's feedback regarding for reward function and scalarization scaing factors (Sec 3.1, Appenix D.1), why HANSOME is not replacement but complement to route planning of Google Maps (, and comparison to LLM, MARL, and V2V research (Appendix A). Again, we appreciate the reviewer's insightful feedback which helps a lot for us to improve the presentation and clarity.
>
> ## Response to Reviewer Concern
> **Concern**: “Communication is not mandatory and the method can still achieve comparable performance with perturbations.”
>
> We clarify that:
> 1. **HANSOME does not mandate communications**, as it is more practical to assume that some agents do not support communications while others may use communication prior to standardization in real-world mixed autonomy settings.
> 2. **Semantic communication is beneficial** for traffic safety and efficiency, and HANSOME is designed to take advantage of this when semantic communication is available.
>
> Our experiments take both scenarios into account.
>
> ### Performance Comparison
> In response to the reviewer’s observation that **“the method can still achieve comparable performance”** when training with disabled intention sharing, we reorganize the performance comparison. The results show an **18%-34% reduction in collision rates** relative to the benchmark using visual only. Efficiency is also improved by **11%-20%**, indicating the development of more confident policies.
>
> ### Collision Rate Comparison of Different Communication Settings
>
> | **Communications**                   | **LeftTurn**       | **RightTurn**      |
> |--------------------------------------|--------------------|--------------------|
> | **visual only**                 | 16.94% ± 4.67%     | 8.38% ± 3.04%      |
> | **visual + intention sharing (HANSOME)** | 13.89% ± 3.21%     | 5.52% ± 0.45%      |
> | **Improvements (HANSOME over visual only)** | **18.00%**         | **34.13%**         |
>
> ### Norm. Speed Comparison of Different Communication Settings
>
> | **Communications**             | **LeftTurn**       | **RightTurn**      |
> |--------------------------------------|--------------------|--------------------|
> | **visual only**                   | 0.50 ± 0.01        | 0.64 ± 0.04        |
> | **visual + intention (HANSOME)**  | 0.60 ± 0.01        | 0.72 ± 0.07        |
> | **Improvements (HANSOME over visual only)** | **20.00%**         | **11.11%**         |

---

> > ### Author Response · Authors · 2024-12-03
> > **Response to Reviewer xCkR**
> >
> > We thank reviewer xCkR for the insightful questions and thorough review during the discussion period. We are glad to have the opportunity to provide further clarification in our responses and enhance the clarity of our paper in the revised version based on your feedback. Specifically,
> >
> > - We re-organized Table 3 to highlight the benefits of semantic communications, addressing your questions about their necessity.
> > - We provided justifications for heuristic neighbor selection and reward design, which are common practices in MARL and RL literature. Additionally, our revision provided an analogy of AdaSMO's scalarization hyper-parameters to adaptive learning rates and discussed why AdaSMO is more intuitive and how it can be robust to different hyper-parameters (from line 1143).
> > - We added new baseline experiments on LeftTurn, RightTurn, and a new task, Roundabout.
> > - We clarified that HANSOME is not a replacement but a complement to Google Maps route planning (line 194).
> > - We added comparisons to MARL, HRL, LLMs, and other V2V works in the Related Works section and Appendix A.
> >
> > **We are happy to answer any further questions on the last day of the discussion phase. We hope our responses and revisions have addressed the remaining concerns. If so, we kindly request the reviewer to consider revising the score before the discussion period ends today.**

---

### Official Review · Reviewer_dXX5 · 2024-11-04

**Soundness:** 3
**Presentation:** 3
**Contribution:** 2
**Rating:** 5
**Confidence:** 2

**Summary:**

"World-Model Based Hierarchical Planning with Semantic Communications for Autonomous Driving" introduces HANSOME (Hierarchical Autonomous Navigation with Semantic Communication), a framework designed to improve autonomous driving using a world-model (WM) approach. HANSOME leverages hierarchical reinforcement learning (HRL) to manage complex, multi-agent driving scenarios by dividing decision-making into high-level intentions and low-level actions. This approach mirrors human driving strategies, where higher-level intentions like lane changes are communicated to other vehicles, while lower-level controls (e.g., acceleration) execute these decisions.

The contributions are summarized as follows:
1) HANSOME has a hierarchical planning strategy where the higher-level policy generates and shares semantic intentions in the form of text to guide the lower-level policy which in turn decides specific controls.

2) hierarchical training as a multi-objective optimization problem and devise AdaSMO to dynamically balance learning of two-level policies

3) Exhasutive experimentation to show where current state-of-the-art WM-based RL methods may fail, and show AdaSMO’s effectiveness in training a good hierarchical planning strategy

**Strengths:**

1) The paper is well written and easy to follow, with appropriate diagrams to help readers understand the complex two level policy design
2) THe related work sections broadly offers a good overview of the world modeling literature
3) There is a lot of technical contribution in terms of both coming up with the two level policy approach to RL and as well design the adasMO objective to optimize the hierarchical policy planning.
4) Exhasutive experimentation and ablation allow the reader to understand the contibution of each of the proposed novelties.

**Weaknesses:**

1) I struggle to find what is the real world application of such a design where each agent needs to communicate policies with other agents to make progress? Are they limited to simulation or a pre training world modeling task to later be applied to a real world planner distribution where all agents are not controlled by a uniform policy? If the later, then the paper should include some analysis of such an adaptation otherwise it is unclear how effective the setup is for such a design. If the former, more exhasutive interactive agent analysis on other publically  available benchmarks must be provided to ascertain technical competitiveness of the proposed methodology.

2) the experimentation section is weak as it only compares to a corner of the planning research world, one one particular dataset.
More exhasutive evaluation of ablation of the choices made would make it clear what are the true contributions of this work.

**Questions:**

1) I struggle to find what is the real world application of such a design where each agent needs to communicate policies with other agents to make progress? Are they limited to simulation or a pre training world modeling task to later be applied to a real world planner distribution where all agents are not controlled by a uniform policy? If the later, then the paper should include some analysis of such an adaptation otherwise it is unclear how effective the setup is for such a design. If the former, more exhasutive interactive agent analysis on other publically  available benchmarks must be provided to ascertain technical competitiveness of the proposed methodology.

---

> ### Author Response · Authors · 2024-11-23
> **Responses to reviewer dXX5**
>
> **Updated this response on Nov 27 with the paper revision**
>
> We thank the reviewer for the feedback. Here are our responses to the main concerns.
>
> ## [W1] Does HANSOME needs communications to make progress? Or is it applied to real world distributions where all agents are not controlled by a uniform policy?
>
> Thank you for the question regarding the application of HANSOME.
>
> ## Communication is not mandatory but it improves safety and efficiency.
>
> As discussed from lines 86-90, HANSOME does not mandate communications to make progress. Shared intentions are an optional input to our cross-modal encoder-decoder. As stated in the paper, shared intentions help the encoder-decoder predict the future trajectory of background vehicles. In absence of such information, the prediction just becomes less accurate, but HANSOME can still make planning decisions based on ego vehicles’ own observations. It is similar to visual information sharing, which contributes to a more complete BEV in case some vehicles are blocked by others.
>
> To further testify to this point in response to the reviewer’s comment, we add another set of experiments to demonstrate HANSOME’s robustness to environments where intention communications are unavailable.  The models are all trained with around 500k steps budget with the same model configurations/hyper-parameters.
>
> | **Communications**         | **LeftTurn**       | **RightTurn**       |
> |-----------------------------------|--------------------|---------------------|
> | w/ visual only     | 82.21% ± 6.94%     | 91.62% ± 3.04% |
> | w/ visual + intention (HANSOME)            | 85.19% ± 4.14%     | 94.27% ± 0.63%      |
>
> ### Handling Non-Uniform Policies.
>
> The communication protocols in HANSOME are agnostic to agent models or policies since agents **do not need to communicate their full policies but rather share their intentions (such as “left turn”)**. As the messages do not involve any agent-specific representations, HANSOME does not impose any restrictions on the lower-level policy. Therefore, our setup is practical in the environment with heterogeneous agents. Furthermore, we point out that during the training/testing, background vehicles include **1) rule-based drivers, including aggressive drivers** (line 474-476, line 536-538) **2) other HANSOME agents** (as discussed in Appendix B Ego-Centric Learning), which have non-uniform policies. Our experiment show that HANSOME is capable of navigating through these environments with mixed policies.
>
> ### Real-world Application with Heterogeneous Agents
>
> HANSOME supports heterogeneous agents by using semantic language as a universal communication interface, unlike conventional multi-agent RL approaches [2, 3, 4] that require homogeneous agents with shared latent spaces. Our experiment results demonstrate that this semantic communication significantly improves traffic safety and efficiency.
> While V2V communication protocols [6, 7, 8, 9] provide the technical foundation for inter-vehicle communication, our contribution focuses on demonstrating that semantic language offers a simple yet effective communication paradigm for heterogeneous agents with different underlying policies.
>
> ## [W2] Comparison experiments.
>
> We clarify that our work focuses on world model based planning and we compare our proposed method with SOTA methods such as DreamerV3, Director [10, 11, 12].
>
> We discussed the selection of baselines and tasks from line 396 and line 409, and discussed how prior works customized their tasks for evaluation. For example, some works have the agent drive to maximize total rewards within 1000 steps [11, 13] in CARLA town03. Our work focuses on multi-agent communications (different from the works on general robotics planning or LLM based planning), and it is necessary for us to design and develop closed-loop tasks that rigorously evaluate the impact of semantic communications. But these tasks are not readily available. In this regard, we have meticulously designed challenging tasks with dense traffic and challenging objectives in a highly interactive environment CARLA. The ego vehicle has to find proper timing to merge lanes, or take turns, or flexibly bypass obstacles ahead, avoid collision, to complete tasks.
>
> We hope the clarifications can address the reviewer’s concerns and we are willing to address any further questions.

---

> > ### Author Response · Authors · 2024-11-23
> > **References**
> >
> > [1] Zhu, Changxi, Mehdi Dastani, and Shihan Wang. "A survey of multi-agent reinforcement learning with communication." arXiv preprint arXiv:2203.08975 1 (2022).
> >
> > [2] Kim, W., Park, J., & Sung, Y. (2020, April). Communication in multi-agent reinforcement learning: Intention sharing. In International conference on learning representations.
> >
> > [3] Xie, A., Losey, D., Tolsma, R., Finn, C., & Sadigh, D. (2021, October). Learning latent representations to influence multi-agent interaction. In Conference on robot learning (pp. 575-588). PMLR.
> >
> > [4] Zhu, C., Dastani, M., & Wang, S. (2022). A survey of multi-agent reinforcement learning with communication. arXiv preprint arXiv:2203.08975.
> >
> > [5] Zhang, Kaiqing, Zhuoran Yang, and Tamer Başar. "Multi-agent reinforcement learning: A selective overview of theories and algorithms." Handbook of reinforcement learning and control (2021): 321-384.
> >
> > [6] Wang, H. M., Avedisov, S. S., Altintas, O., & Orosz, G. (2023, April). Evaluating intent sharing communication using real connected vehicles. In 2023 IEEE Vehicular Networking Conference (VNC) (pp. 69-72). IEEE.
> >
> > [7] Wang, H. M., Avedisov, S. S., Altintas, O., & Orosz, G. (2024). Intent sharing in cooperative maneuvering: Theory and experimental evaluation. IEEE Transactions on Intelligent Transportation Systems.
> >
> > [8] Wang, H. M., Avedisov, S. S., Altintas, O., & Orosz, G. (2023, June). Experimental validation of intent sharing in cooperative maneuvering. In 2023 IEEE Intelligent Vehicles Symposium (IV) (pp. 1-6). IEEE.
> >
> > [9] Runsheng Xu, Hao Xiang, Zhengzhong Tu, Xin Xia, Ming-Hsuan Yang, and Jiaqi Ma. V2x-vit: Vehicle-to-everything cooperative perception with vision transformer. In European conference on computer vision, pages 107–124. Springer, 2022.
> >
> > [10] Hu, Anthony, et al. "Model-based imitation learning for urban driving." Advances in Neural Information Processing Systems 35 (2022): 20703-20716.
> >
> > [11] Gao, Z., Mu, Y., Chen, C., Duan, J., Luo, P., Lu, Y., & Li, S. E. (2024). Enhance sample efficiency and robustness of end-to-end urban autonomous driving via semantic masked world model. IEEE Transactions on Intelligent
> >
> > [12] Li, Qifeng, et al. "Think2drive: Efficient reinforcement learning by thinking in latent world model for quasi-realistic autonomous driving (in carla-v2)." arXiv preprint arXiv:2402.16720 (2024).
> >
> > [13] Pan, Minting, et al. "Iso-dream: Isolating and leveraging noncontrollable visual dynamics in world models." Advances in neural information processing systems 35 (2022): 23178-23191.

---

### Meta-Review · Area_Chair_ED5t · 2024-12-22

**Metareview:**

The paper proposes a world model based hierarchical planning for autonomous driving. The proposed framework, namely HANSOME, is an hierarchical design to leverage both low-level policy with high-level semantics. All reviewers send out constructive feedback; the main concerns are:

- Motivation as to why the communications are necessary. What's the real-world application.
- Insufficient experiments. Some baselines are very simple. How to handle unpredictable driving behaviors.
- Heuristic design in HANSOME

The paper is clearly written and easy to follow. To sum up, this paper slightly falls below the acceptance bar for ICLR. The major concerns are, the paper might be better polished with a focused "gist" of the technical novelty, rather than splited out in several aspects (Communications, V2X, AdsSMO). If the paper do view the communication part as important, it could be better to incorporate some baselines from V2X domain.

**Additional Comments On Reviewer Discussion:**

Reveiwers acknowledge the feedback and responded accordingly. After rebuttal, only one review give positive rating. The other reviewer's comments mainly concentrate on (a) whether the motivation / real-world applications remain / novelty. (b) lack of comparision experiments.

While the writing is clear, the current version of the manuscript indeed is not ready for acceptance. Authors are strongly encouraged to polish the storytelling and revise the experiments. It would be greatly enhanced to submit at future venues.

---

### Decision · Program_Chairs · 2025-01-22

Reject